

# Description of the uEMEP_v5 downscaling approach for the EMEP MSC-W chemistry transport model

Bruce Rolstad Denby[1], Michael Gauss[1], Peter Wind[1,2], Qing Mu[1], Eivind Grøtting Wærsted[1], Hilde Fagerli[1], Alvaro Valdebenito[1], Heiko Klein[1]

[1] The Norwegian Meteorological Institute, Henrik Mohns Plass 1, 0313, Oslo, Norway
[2] Department of Chemistry, UiT - The Arctic University of Norway, N-9037 Tromsø, Norway

*Correspondence to*: Bruce Rolstad Denby (brucerd@met.no)

**Abstract.** A description of the new air quality downscaling model uEMEP and its combination with the EMEP MSC-W chemistry transport model is presented. uEMEP is based on well known Gaussian modelling principles. The uniqueness of the system is in its combination with the EMEP MSC-W model and the 'local fraction' calculation contained within it. This allows the uEMEP model to be imbedded in the EMEP MSC-W model and downscaling can be carried out anywhere within the EMEP model domain, without any double counting of emissions, if appropriate proxy data is available that describe the spatial distribution of the emissions. This makes the model suitable for high resolution calculations, down to 50 m, over entire countries. An example application, the Norwegian air quality forecasting and assessment system, is described where the entire country is modelled at a resolution of between 250 and 50 m. The model is validated against all available monitoring data, including traffic sites, in Norway. The results of the validation show good results for $NO_2$, which has the best known emissions, and moderately good for $PM_{10}$ and $PM_{2.5}$. In Norway the largest contributor to PM, even in cities, is long range transport followed by road dust and domestic heating emissions. These contributors to PM are more difficult to quantify than $NO_x$ exhaust emission from traffic, which is the major contributor to $NO_2$ concentrations. In addition to the validation results a number of verification and sensitivity results are summarised. One verification showed that single annual mean calculations with a rotationally symmetric dispersion kernel give very similar results to the average of an entire year of hourly calculations, reducing the run time for annual means by four orders of magnitude. The uEMEP model, in combination with EMEP MSC-W model, provides a new tool for assessing local scale concentrations and exposure over large regions in a consistent and homogenous way and is suitable for large scale policy applications.





# 1. Introduction

The EMEP MSC-W model is a chemistry transport model, which has been developed by the Meteorological Synthesizing
Centre - West (MSC-W) of EMEP, the European Monitoring and Evaluation Programme under the UN Convention on Long-
range Transboundary Air pollution (LRTAP). It is documented in Simpson et al. (2012) and has been used for many years
mainly for regional but also for global applications. The aim of uEMEP (urban EMEP) is to further extend the application of
the EMEP MSC-W chemical transport model down to near street level modelling. The downscaling methodology builds on
classical Gaussian plume modelling and is integrated with the EMEP MSC-W models physical parameterisations and emission
data in such a way as to provide a consistent model description from regional to local scales.  Unlike other urban scale models
uEMEP is intended not just to achieve local scale modelling for one individual city or area but to provide local scale modelling
over entire countries or continents, providing high resolution modelling over large areas and allowing air quality assessment
and exposure calculations at high resolution everywhere.

Air quality modelling is often separated into global, regional, urban and local scales. In this context local refers to the ability
of the model not just to represent urban background concentrations but also to represent street level concentrations. There are
a number of models that span the global or regional scale where grid resolutions down to 4-10 km have been achieved, e.g.
EMEP MSC-W (Simpson et al., 2012; Werner et al., 2018), CHIMERE (Menut et al., 2013) , SILAM (Sofiev et al., 2015),
LOTOS-EUROS (Kranenburg et al., 2013), MATCH (Andersson et al., 2007) and CMAQ (Appel et al., 2017). There are a
number of Gaussian modelling systems that cover the urban and local scales over limited areas, usually individual cities, e.g.
ADMS (Stocker et al., 2012) and AERMOD (Cimorelli et al., 2004). In addition there are some limited area models that
combine Eulerian and Gaussian plume type models in a single system, e.g. Karamchandani  et al. (2009), Kim et al. (2018)
and Karl et al. (2019). If the full model cascade is to be achieved, such as the THOR forecast system in Denmark (Brandt et.al.,
2001), then this requires linking different model systems together to achieve high resolution calculations in a limited area. An
alternative approach to achieving high resolution concentration fields over large regions without the use of CTMs are land use
regression methods (e.g. Vizcaino and Lavalle; 2018), however their lack of underlying physics do not make them useful for
planning or policy applications.

Earlier work on the downscaling of CTM models over large regions include the population covariance correction factor from
Denby et al. (2011) and the dispersion kernel methods from Theobald et al. (2016) and Maiheu et al. (2017). There are
similarities between uEMEP and these last two studies as both use Gaussian models for the downscaling. The major difference
between uEMEP and these two Gaussian kernel methods is that uEMEP can be applied on hourly data, as well as annual data,
and that uEMEP is integrated with the 'local fraction' scheme in EMEP MSC-W (Wind et al., 2020) that avoids double
counting of emissions in a consistent manner.




In this paper we provide an overall description of the uEMEP methodology and how it is combined with the 'local fraction' scheme in EMEP MSC-W (Sect. 2). The uEMEP model physical parameterisations are then given in Sect. 3. In Sect. 4 an application example of the methodology, the Norwegian air quality forecasting service, is described. Validation of the modelling system against all Norwegian monitoring data is presented in Sect. 5 together with a summary of verification and sensitivity studies. Various aspects of the modelling are discussed in Sect. 6 and conclusions made in Sect. 7. Supplementary material providing more detailed information on the parameterisations, validation and verification is also aprovided.

## 2. Methodology

In this section we describe the concepts and methodologies for the application of the coupled modelling system uEMEP and EMEP MSC-W.

### 2.1 Overall concept and methodology

uEMEP provides a consistent methodology for redistributing/downscaling gridded CTM (Chemical Transport Model) emissions and concentrations, in this case from the EMEP MSC-W model, to higher resolution sub-grids within the CTM grids. Proxy data, that represent the spatial distribution of the emissions, are used to redistribute emissions to sub-grids. Concentrations are then recalculated at the sub-grid level, using a Gaussian model, whilst removing the local contribution of the CTM at these sub-grids but keeping the non-local contributions. This procedure consistently avoids double counting of emissions.

Throughout this paper we refer to the downscaling grids in uEMEP as 'sub-grids'. These may be any size but current applications use a range of between 25 and 250 m. When referring to the EMEP MSC-W model we use the term 'grid'. This may also vary dependent on the application but is usually in the range of 2 to 15 km. The term 'local' is also used. Local, in regard to EMEP, means the local EMEP grids, so terms such as' local fraction' refer to a particular grid and the other EMEP grids in the 'local region', for example within a range of 5 x 5 EMEP grids. When referring to 'local' in uEMEP we also refer to sub-grid contributions from within this local EMEP region. This is visualised in Fig. 1. 'Non-local', in regard to uEMEP, refers to any contribution that is not downscaled or calculated with uEMEP, usually contributions from outside the local EMEP region but these can also be other source sectors not accounted for by uEMEP. We will refer to concentrations provided by the EMEP MSC-W model simply as EMEP concentrations.

uEMEP can be run using two downscaling methods, both of which make use of Gaussian dispersion modelling to describe the redistribution of concentrations at high resolution. Both methods make use of the recently developed 'local fraction' functionality in the EMEP model (Sect. 2.3; Wind et al., 2020) to avoid double counting of emissions and to allow near seamless integration of the two models. The two downscaling methods are:





1. **Emission redistribution**: Redistribution of the EMEP gridded emissions using emission proxy data to sub-grids and calculation of concentrations through Gaussian modelling, with removal of the locally emitted EMEP concentrations.


2. **Independent emissions:** Independent high resolution emission data on sub-grids and calculation of the concentrations through Gaussian modelling, with removal of the locally emitted EMEP source contributions.

In addition calculations can be made on either long term mean emissions, using a rotationally symmetric dispersion kernel (Sect. 3.2), or on hourly emissions, using a slender plume Gaussian dispersion model (Sect. 3.1).


uEMEP is applied to the primary emissions of $PM_{10}$, $PM_{2.5}$ and $NO_x$ and does not involve any complex chemistry or secondary formation of particles. The concentrations of $NO_2$ and $O_3$ are calculated with uEMEP using a simplified chemistry scheme (Sect. 3.4 and 3.5).

## 2.2 Sub-grid calculation method

The choice of downscaling method will depend on the quality of the high resolution proxy or emission data available, whether the calculation will be made on hourly or annual data and on the EMEP model resolution. The first downscaling method, emission redistribution, will be applied when only approximate proxy data for emissions are available, which will be the case in many large scale applications. Examples of such proxy data may be population density, road network data or land use data. The second downscaling method, independent emissions, is suitable for when high quality emission data is available that is

consistent between uEMEP and EMEP. When the gridded emission data is entirely consistent with the proxy data, i.e. the proxy data are given as emissions and are aggregated to the CTM grid emissions, then the two methods are equivalent.

The total sub-grid concentrations $C_{SG}(i,j)$ at sub-grid indexes *(i,j)* are calculated by adding the local Gaussian concentration $C_{SG,local}(i,j)$ and the non-local part of the EMEP grid concentration $C_{SG,nonlocal}(i,j)$ and is written simply as


$$C_{SG}(i,j) = C_{SG,local}(i,j) + C_{SG,nonlocal}(i,j) \qquad (1)$$

where we use the subscript notation '*SG*' to denote any sub-grid value and in further equations the subscript '*G*' to indicate any EMEP grid value. $C_{SG,local}(i,j)$ is determined directly from the sub-grid dispersion calculation


$$C_{SG,local}(i,j) = \sum_{i'=1}^{n_x} \sum_{j'=1}^{n_y} \frac{E_{SG,local}(i',j')}{U(i,j,i',j')} \cdot I\big(r(i,j,i',j'), h(i',j'), z(i,j)\big) \qquad (2)$$





Here $E_{SG,local}$ is the emission attributed to each sub-grid and $U$ is the wind speed, which in uEMEP is dependent on both the source and receptor sub-grid values (Sect. 3.1). $n_x$ and $n_y$ represent the extent of the sub-grid calculation window. The function

$I(r,h,z)$ is the dispersion intensity function (Sect. 3.1) that determines the dispersion of the emission source $E_{SG,local}$ with the horizontal spatial vector $r(i,j,i',j')$ between the receptor grid points $(i,j)$ and the source grid points $(i',j')$ at height $z(i,j)$. The source height $h(i',j')$ is also specified. The contribution from every proxy emission sub-grid $(i',j')$, within the defined sub-grid calculation window $(n_x, n_y)$ is calculated and summed at the receptor sub-grid $(i,j)$ centered within sub-grid calculation window, see Fig. 1.


When using the emission redistribution method then $E_{SG,local}$ is calculated using the EMEP grid emissions $E_G(I,J)$ and the proxy data for emissions, $P_{emission}$. $P_{emission}$ is normalised within the EMEP grid in the following way to determine the sub-grid emission $E_{SG,local}$

$$E_{SG,local}(i',j') = E_G(I,J)\frac{P_{emission}(i',j')}{\sum_{i'=1}^{n_x}\sum_{j'=1}^{n_y}P_{emission}(i',j')} \qquad (3)$$

When using the independent emission method then the local sub-grid emissions $E_{SG,local}$ are specified directly.

## 2.3 Calculation of the non-local contribution from EMEP

The term $C_{SG,nonlocal}(i,j)$ in Eq. (1) is the non-local contribution from the EMEP grid at the specific sub-grid $(i,j)$. Though this is based on the non-local contribution provided by EMEP at grids $(I,J)$ interpolation due to the moving window (Sect. 2.4) surrounding each receptor sub-grid means that non-local contributions are specified at the sub-grid level. The gridded non-local contribution $C_{G,nonlocal}(I,J)$ is derived from the 'local fraction' calculation in EMEP. The methodology is described completely in Wind et al. (2020) but the essential elements are reproduced here.


The local fraction methodology corresponds to a tagging method, where pollutants from different origins are tagged and stored individually. In this case the tagging occurs relative to the surrounding grid cells of any individual grid. This means that emissions from any grid cell are tagged and followed through the various model processes out to neighbouring grid cells. It is generally not computationally possible, or in this application necessary, to follow all grid cell contributions to all other grids

within the EMEP model domain. The local fractions are then limited to neighbour cells. In Wind et al. (2020) the local fraction region extent $(n_{lf})$ was tested up to a 161 x 161 EMEP grids on low resolution EMEP runs for Europe. Generally 21 x 21 EMEP grids were found to be computationally and memory efficient. In the uEMEP application the local fraction region needs only be as large as the uEMEP calculation window, i.e. the allowed distance from the receptor sub-grid to the emission sub-grids.





In the forecasting application discussed in Sect. 0 this requires only a 5 x 5 EMEP grid local fraction region. Sensitivity to the

size of this region is discussed in Sect. 5.2.

With use of the local fraction then the local ($C_{G,local}$) and non-local ($C_{G,nonlocal}$) contributions from any particular primary

pollutant in EMEP is given by the sum of the local sources ($s = 1$ to $n_{source}$) and the non-local contribution determined where

$$C_{G,local}(I,J,s) = LF(I,J,s)\,C_G(I,J) \tag{4}$$

$$C_{G,nonlocal}(I,J) = C_G(I,J) - \left\{\sum_{s=1}^{n_{source}} C_{G,local}(I,J,s)\right\} \tag{5}$$

Note that in Wind et al. (2020) $C_{G,local}$ and $C_G$ are termed *LP* (local pollutant) and *TP* (total pollutant) respectively. This change

is for compatibility with the notation used for the uEMEP application.

**2.4 Moving window calculation of local and non-local EMEP contributions**

When determining the local and non-local EMEP contribution at any uEMEP sub-grid receptor then a moving window

methodology is applied. The aim of the moving window calculation is to represent as well as possible the local and non-local

EMEP contributions at any one sub-grid, in effect creating an EMEP grid that is centred on the receptor sub-grid. The moving

window is centred on the receptor sub-grid ($i,j$) and its size is specified by the number of EMEP grids it covers ($n_{mw}$, $n_{mw}$). The

moving window region is the same as the uEMEP calculation window in extent, which is also defined by the number of sub-

grids ($n_x$, $n_y$), Sect. 2.2. $n_{mw}$ is given by the user but it must not be larger than the area covered by the EMEP local fraction

region ($n_{lf}$), i.e. $n_{mw} \leq n_{lf} - 1$. Fig. 1 shows an example where $n_{lf} = 5$ and $n_{mw} = 4$.

Since we need to account for all source contributions from EMEP within the moving window and since the sub-grids are not

centred in the middle of the EMEP grids then the local contribution from the EMEP grids for any particular source sector *s*

can be written as

$$C_{G,local}(i,j,s) = \sum_{I'=I-n_{mw}/2}^{I+n_{mw}/2} \sum_{J'=J-n_{mw}/2}^{J+n_{mw}/2} C_{G,local}(I',J',s) \cdot w(i,j,I',J',s) \tag{6}$$


Here the weighting variable *w(i,j,I',J',s)* refers to the weighting of the EMEP grid relative to the receptor sub-grid. For EMEP

grids entirely within the moving window then this weighting will be unity, but for EMEP grids only partially within the moving

window this weighting will be less than unity as part of that EMEP grid will also contribute to the non-local concentrations.





There are two methods implemented in uEMEP for specifying these weights. The simplest and most often used is area
weighting where only the area fraction of the EMEP grid that is within the moving window for that particular receptor sub-
grid is included in the local contribution. This is illustrated in Fig. 1 and is usually sufficient for the calculation, especially
when the number of EMEP grids covered by the moving window is larger than 3 x 3. Mathematically the area weighting, *wa,*
can be written as


$$wa(i,j,I',J') = \frac{\{a(i,j) \cap A(I',J')\}}{A(I',J')} \tag{7}$$

where $A(I',J')$ is the area and position of each EMEP grid, $a(i,j)$ is the area and position of the moving window centred at the
receptor sub-grid point *(i,j)* and $a(i,j) \cap A(I',J')$ is the overlapping area of these two regions. For the case where $n_{mw} = 1$ then
this area weighting is equivalent to a bilinear interpolation of the surround EMEP grids. Area weighting is not dependent on
the source.

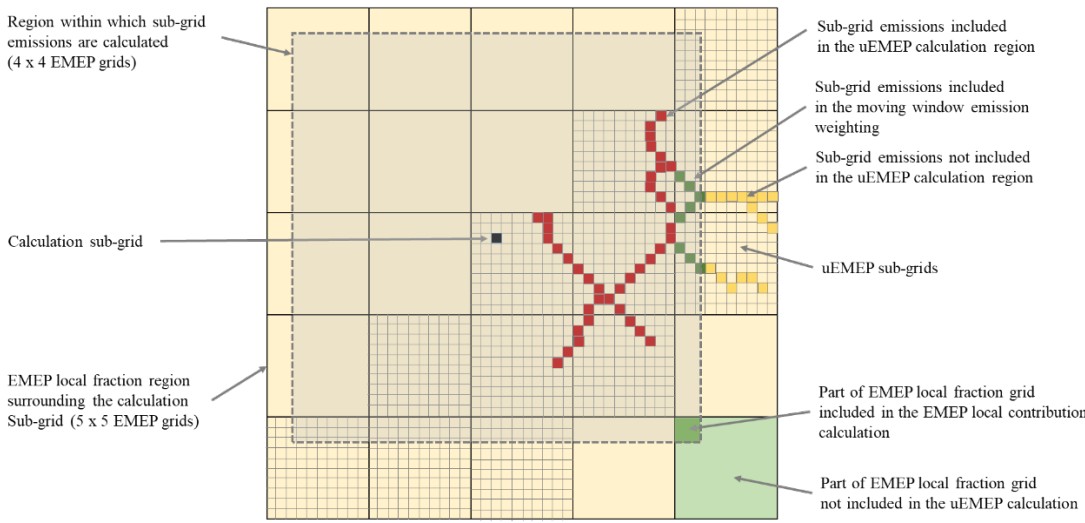

Schematic representation of the uEMEP moving window region

**Figure 1. Schematic representation of the moving window region. It shows the regions used for uEMEP calculations and the area**
**and emission weighting selection used to determine the local and non-local EMEP contributions at the calculation (receptor) sub-**
**grid. The extent of the sub-grids is only partially shown.**

When the moving window only covers a limited number of EMEP grids and when high resolution emission data is used that
is compatible with the EMEP grid emissions, then this weighting can also be based on the high resolution emission data itself.
This better represents the moving window concept because it reflects the effect of moving the EMEP grid to be centred on the





receptor sub-grid in a more realistic way. In this case the emission weighting term (*we*) on the edge of the moving window will be determined by the fraction of the total sub-grid emissions within the moving window and within the EMEP grid, instead of the area. This can be written as


$$we(i,j,I',J',s) = \frac{\sum e(i,j,I',J',s): \in \{a(i,j) \cap A(I',J')\}}{\sum e(i,j,I',J',s): \in \{A(I',J')\}} \tag{8}$$

where the numerator is the sum of the emissions within the intersection of *a(i,j)* and *A(I',J')* and the denominator is the sum of the emissions within *A(I',J')*. The resulting total concentration, using this method, may be higher or lower than the original

EMEP concentrations because it reflects the impact of moving the EMEP grid in space. Due to this it is not possible to simply subtract the local EMEP contribution from the total to get the non-local EMEP contribution when using this method.

To address this the non-local EMEP contribution is also calculated using the moving window with Eq. (9). The first term is the non-local contribution for a particular source and is calculated with the area weighting distribution, as non-local

contributions do not have any associated emission. An additional correction term, second term in Eq. (9), that accounts for the non-local contributions from other local sources must be included. This is done by subtracting the local source contribution from the calculated non-local value.

$$C_{G,nonlocal}(i,j,s) = \sum_{I'=I-n_{mw}}^{I+n_{mw}} \sum_{J'=J-n_{mw}}^{J+n_{mw}} C_{G,nonlocal}(I',J',s) \cdot wa(i,j,I',J') -$$

$$\sum_{I'=I-n_{mw}\ (I'\neq I)}^{I+n_{mw}} \sum_{J'=J-n_{mw}\ (J'\neq J)}^{J+n_{mw}} \begin{pmatrix} C_{G,local}(I' \rightarrow I, J' \rightarrow J, s) \cdot w(i,j,I',J',s) \\ + C_{G,local}(I \rightarrow I', J \rightarrow J', s) \cdot w(i,j,I,J,s) \end{pmatrix} \tag{9}$$

In Eq. (9) the weighting term *w* represents either the emission (*we*) or the area (*wa*) weighting, depending on the choice of weighting method.

These local and non-local calculations are carried out for each emission source individually so the non-local contribution is also dependent on source. The total local contribution $C_{G,local}$ is given by the sum of all the local source contributions, as in the last term of Eq. (5), and the final non-local contribution at each sub-grid is calculated using

$$C_{G,nonlocal}(i,j) = \sum_{s=1}^{n_{source}} \left\{ \frac{C_G(i,j,s)}{n_{source}} - C_{G,local}(i,j,s) \right\} \tag{10}$$


In the case of area weighting, where the sum of local and non-local is the same as the original EMEP total concentration, then the first term in the summation is equivalent to the original EMEP concentration without summation. The method is illustrated in one dimension in Fig. 2.


The calculation based on emission weighting is computationally more expensive than the area weighting and is only used when necessary and appropriate, e.g. when $n_{mw} = 1$ and when sub-grid and grid emissions are consistent with each other.

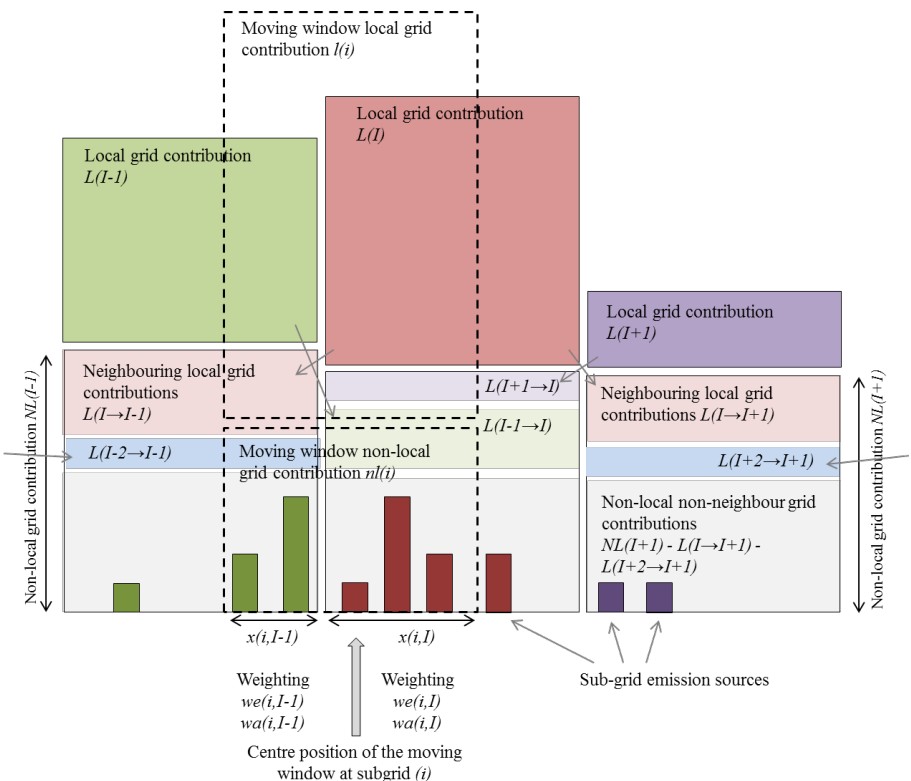

**Figure 2. Illustration of the moving window interpolation method employed in uEMEP. Shown is the 1D visualisation of the 2D method described in Equ. 6 - 10 for $n_{mw} = 1$. For clarity in the figure the terms $C_{G,local}$ and $C_{Gnonlocal}$ are written as $L$ and $NL$ respectively.**

## 3. uEMEP model process description and parameterisation

In this section uEMEP process parameterisations are described. In regard to the dispersion modelling uEMEP is intended to
integrate closely with EMEP. To enable this dispersion schemes based on parameterisations used in EMEP have been implemented. In the supplementary material additional equations (Sect. S1-S3) are provided and a number of optional additional parameterisations are also described (Sect. S4).





### 3.1 Sub-grid Gaussian dispersion modelling for hourly calculations

A standard Gaussian narrow plume dispersion model formulation, e.g. Seinfeld and Pandis (1998), is used in the sub-grid
dispersion calculations with multiple reflections from the surface ($z=0$) and boundary layer height ($z=H$). Generically the
Gaussian plume calculation can be written as

$$C(x,y,z) = \frac{Q}{U} I(x,y,z) \qquad (11)$$

where for the sake of clarity we have dropped references to sub-grid indexes as given in Sect. 2 and use coordinates instead of
indices. Here $C$ is the concentration, $Q$ is the emission source strength and $I$ is the plume intensity given by

$$I(x,y,z) = \frac{1}{2\pi\sigma_y\sigma_z} exp\left(\frac{-y^2}{2\sigma_y{}^2}\right) \sum_{i=1}^{i=6}\left\{exp\left(\frac{-(z-h_i)^2}{2\sigma_z{}^2}\right)\right\} \qquad (12)$$

Here $h_i$ represents the plume emission height and five additional virtual plume emission heights after single and double
reflections from the surface and boundary layer top ($H$) given by

$$h_i = [h_{emis}, -h_{emis}, 2H - h_{emis}, 2H + h_{emis}, -2H + h_{emis}, -2H - h_{emis}] \qquad (13)$$

For the well mixed plume case, when $\sigma_z$ is of the order of $H$, we define a threshold beyond which the plume concentration is
constant throughout the boundary layer. This is specified to occur when $\sigma_z > 0.9H$ leading to an intensity given by

$$I(x,y,z) = \frac{1}{\sqrt{2\pi}\sigma_y H} exp\left(\frac{-y^2}{2\sigma_y{}^2}\right) \qquad (14)$$

The Gaussian dispersion parameters $\sigma_z$ and $\sigma_y$ used in Eq. (12) may be determined empirically (Smith, 1973; Martin, 1976;
Turner, 1994; Liu et al., 2015) or through a range of methods based on theoretical and semi-empirical considerations (Seinfeld
and Pandis, 1998). Venkatram (1996) also discusses the relationship between empirically and theoretically based dispersion
parameters. Standard Gaussian plume models do not take into account variable vertical profiles of wind speed or diffusivity.
Some non-Gaussian descriptions are available based on the application of power laws to these profiles and the vertical
integration of the diffusion equation (Chaudhry and Meroney, 1973; van Ulden, 1978; Venkatram et al., 2013) but this then
creates the problem of defining power laws that 'fit' varying wind and dispersion profiles over the entire boundary layer.
Instead of this we use the center of mass of the plume ($z_{cm}$) to define the height at which the advective wind speed and eddy
diffusivity ($K_z$) are defined and allow this to vary dependent on the plume travel distance, giving a similar effect to the plume
dispersion as the non-Gaussian vertically integrated derivation. A similar methodology is employed by the OPS model (Sauter





et al, 2018). We then use a combination of eddy diffusivity ($K_z$) vertical profiles, Lagrangian time scales and centre of mass
plume placement, along with initial values $\sigma_{z0}$ and $\sigma_{y0}$, to determine $\sigma_z$ and $\sigma_y$ values. The aim of this combination is to provide
realistic plume dispersion over short distances but to asymptotically approach the same $K_z$ values used in the EMEP model
dispersion scheme over longer distances. In addition the methodology is implementable at all emission heights and takes into
account both surface roughness and atmospheric boundary layer height.


Following methodologies outlined in Seinfeld and Pandis (1998), we describe the dispersion parameters $\sigma_z$ and $\sigma_y$ as a function
of time using

$$\sigma_z(t) = \sigma_{z0} + \sqrt{2K_z(z)t\,f_t} \tag{15a}$$

$$\sigma_y(t) = \sigma_{y0} + \sqrt{2K_y(z)t\,f_t} \tag{15b}$$

where $t$ is the time and $f_t$ is a factor dependent on the Lagrangian integral time scale $\tau_l$ given by

$$f_t = 1 + \left(\frac{\tau_l}{t}exp\left(-\frac{t}{\tau_l}\right) - 1\right) \tag{16}$$


There are many varying methods for calculating the Langrangian integral time scale (Seinfeld and Pandis, 1998; Hanna, 1981;
Venkatram, 1984). We use the formulation from Hanna (1981)

$$\tau_l = 0.6\frac{max(z_{emis}, z_{\tau min})}{u_*} \quad where \ \ z_{\tau min} = 2\,m \tag{17}$$


Time is calculated from the advective velocity

$$t = \frac{max(x_{min}, x)}{U(z)} \tag{18}$$

where $x_{min}$ is half a sub-grid.

In order to be compatible with the EMEP model the same $K_z$ vertical profile parameterization is used in Eq. (15a) that is used
in EMEP (Simpson et al., 2012). This parameterization is provided in the supplementary material, Eq. (S1-S2).

The center of mass of the plume is calculated using the same Gaussian formulation with reflection as given in Eq. (12) by
integrating the plume intensity over the boundary layer height ($H$) using





$$z_{cm} = \frac{\int_0^H z\,I(z)\,dz}{\int_0^H I(z)\,dz} \tag{19}$$

This integral can be analytically solved to give

$$z_{cm} = \frac{\sigma_z}{\sqrt{2\pi}} \sum_{i=1}^{i=6} exp\left(\frac{-h_i{}^2}{2\sigma_z{}^2}\right) - exp\left(\frac{-(H-h_i)^2}{2\sigma_z{}^2}\right) + \frac{h_i}{2}\left(erf\left(\frac{h_i}{\sqrt{2}\sigma_z}\right) + erf\left(\frac{(H-h_i)}{\sqrt{2}\sigma_z}\right)\right) \tag{20}$$

and for the well mixed case where


$$\sigma_z > 0.9H \ \text{ then } z_{cm} = 0.5\,H \tag{21}$$

The vertical wind profile is calculated in a similar way to Gryning et al. (2007), based on decreasing turbulent shear with height.


$$U(z) = \frac{u_{*0}}{\kappa}\left(log\left(\frac{z}{z_0}\right) - \psi_m + \kappa\frac{z}{z_l}\left(1 - \frac{z}{2H}\right) - \frac{z}{H}\left(1 + b\frac{z}{2L}\right)\right) \ for \ L \geq 0$$

$$U(z) = \frac{u_{*0}}{\kappa}\left(log\left(\frac{z}{z_0}\right) - \psi_m + \kappa\frac{z}{z_l}\left(1 - \frac{z}{2H}\right) - \frac{z}{H}\frac{((a\,z-L)\phi_m+L)}{a\,(p+1)}\right) \ for \ L < 0 \tag{22}$$

The stability functions $\psi_m$ and $\phi_m$ are defined in the supplementary material, Eq. (S3-S4), and the assumptions behind the wind

profile derivation are given in Eq. (S5-S8). There is no turning of the wind direction with height. Eq. (22) is used to derive $u_{*0}$, based on modelled 10 m wind speed, boundary layer height and surface roughness length $z_0$. The vertical wind profile is then derived from this.

The average of the plume center of mass height at the receptor point and the emission height, $z_{av} = 0.5\ (z_{cm} + h_{emis})$, is then

used to determine the vertical diffusion $K_z(z_{av})$ as well as the wind speed $U(z_{av})$ for use in Eq. (15) and (18). The entire set of equations, Eq. (15-22) are solved iteratively to obtain the final $\sigma_z$ value at the receptor point. This iteration converges swiftly and generally only two iterations are required.

The horizontal eddy diffusivity $K_y$ is not determined in EMEP so an alternative is required. $K_y$ can be classically related to $K_z$

through the relationship





$$K_y(z) = \frac{\sigma_v(z)^2}{\sigma_w(z)^2} K_z(z) \tag{23}$$

based on the concepts used to define $K$ (Seinfeld and Pandis, 1998). Garratt (1994) provides expressions for the vertical profile
$\sigma_v$ and $\sigma_w$ under unstable conditions where the ratio $\sigma_v/\sigma_w$ is around 1.85 in the surface layer but decreases to 1 in the mixed
layer. Under stable conditions Nieuwstadt (1984) provides local scaling where this ratio is close to 2. For the current application
we choose the ratio $\sigma_v/\sigma_w = 2$ and apply it over the whole boundary layer.

It is also possible within the modelling setup to use the simpler empirical formulations of $\sigma_z$ and $\sigma_y$, as presented in Eq. (24)
and shown in Table 1. This is useful for testing and comparison and necessary when using the rotationally syemtric plume
parameterization, Sect. 3.2. See Seinfeld and Pandis (1998) for a presentation of these.

In Figure 3 we show two example sets of $\sigma_z$ curves for near surface (1 m) and elevated (50 m) releases as calculated with the
$K_z$ methodology for three separate stabilities. For reference the dispersion curves from ASME (American Society of
Mechanical Engineers), Smith (1973) are also shown. These often used dispersion parameters are relevant for one hour
averaging times. The ASME $\sigma_z$ curves are given in Pasquill stability classes and the conversion from their dependency on
Monim-Obhokov length ($L$) and surface roughness ($z_o$) is achieved using the conversion methodology described by Golder
(1972). Parameters used in the calculation of the 3 curves are provided in Table 1.

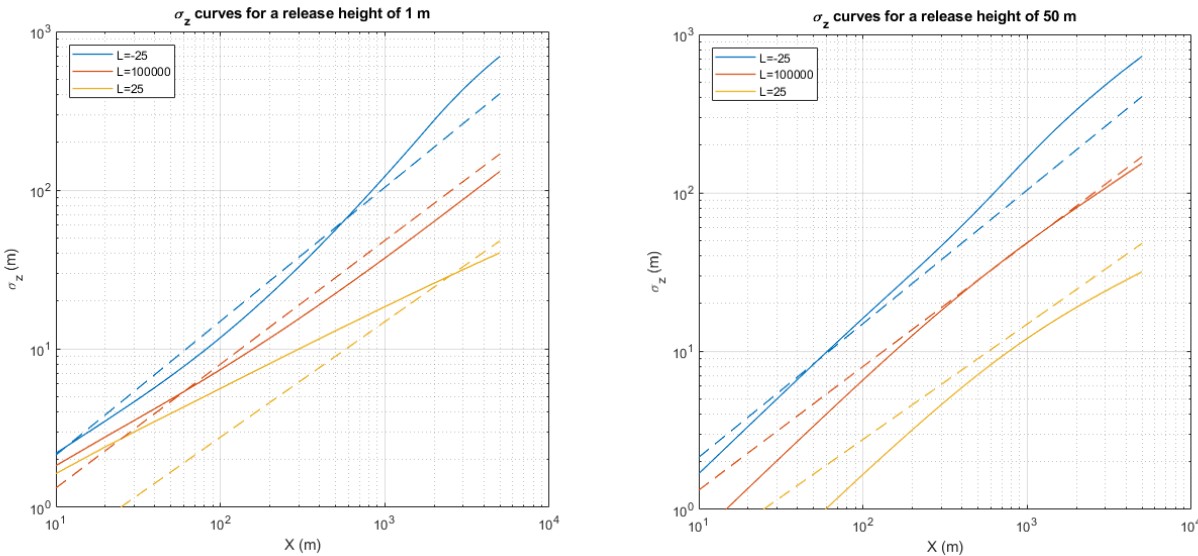


**Figure 3. Comparison of derived $\sigma_z$ curves discussed in the text with standard ASME curves (Smith, 1973) using Eq. (24). To the left a 1 m release and to the right a 50 m release. Three different stability classes, specified by the Monim-Obhukhov lengths ($L$), are**

Unstable





**shown. The $K_z$ method is shown as a solid line and the ASME curves as dashed lines. The ASME curves have no release height or surface roughness dependence but are included as reference. Values of $z_0$=0.5 m, relevant for urban areas, and $\sigma_{z0}$ = 0 are used.**


**Table 1. Parameters used for calculating the curves shown in Fig. 3.**

| Stability | $z_0$ (m) | $H_{bl}$ (m) | $L_{MO}$ (m) | ASME $a_z$ | ASME $b_z$ |
|-----------|-----------|--------------|--------------|------------|------------|
| **Unstable** | 0.5 | 2000 | -25 | 0.401 | 0.844 |
| **Neutral** | 0.5 | 1000 | +100000 | 0.22 | 0.780 |
| **Stable** | 0.5 | 100 | +25 | 0.097 | 0.728 |

### 3.2 Rotationally symmetric Gaussian plume model for annual mean calculations

When applying uEMEP to annual mean emissions a rotationally symmetric Gaussian plume is used. It is possible to derive an

approximate analytical solution to the Gaussian plume equation assuming that wind directions are homogeneously distributed in all directions and that there is no strong dependence of wind speed or stability on wind direction. These conditions are usually not met but it is useful to have such a simplified analytical solution.

The starting point is the Gaussian plume model given in Eq. (12). In this case we do not derive $\sigma_{y,z}$ using the $K_z$ value from

EMEP but apply the commonly used power law formulation in order to derive an analytical solution

$$\sigma_{(y,z)} = \sigma_{0(y,z)} + a_{(y,z)} x^{b(y,z)} \tag{24}$$

Values for the dispersion parameters $a$ and $b$ may be taken from the literature (Seinfeld and Pandis, 1998) but we use the

ASME curves (Smith, 1973) under neutral conditions to specify these.

The rotationally symmetric version of this equation can be derived by rewriting the equation in cylindrical coordinates with appropriate approximations (second order), based on the slender plume assumption, and integrating over all angles. The resulting rotationally symmetric intensity $I_{rot}(r,z)$ as a function of $r$ and $z$ is then written


$$I_{rot}(r,z) = \frac{1}{\pi\sqrt{2\pi}r\varepsilon_z\sqrt{1+B}} \, erf\left(\frac{\pi\sqrt{1+B}}{\sqrt{2}\varepsilon_\theta}\right) \sum_{i=1}^{i=6}\left\{exp\left(\frac{-(z-h_i)^2}{2\varepsilon_z{}^2}\right)\right\} \tag{25}$$

where

$$\varepsilon_z = \sigma_{0z} + a_z r^{b_z} \tag{26a}$$
$$\varepsilon_\theta = \frac{1}{r}\left(\sigma_{0y} + a_y r^{b_y}\right) \tag{26b}$$





$$B = -\varepsilon_\theta{}^2 \left( \frac{b_z(\varepsilon_z - \sigma_{0z})}{r\varepsilon_\theta} + \frac{b_y(r\varepsilon_\theta - \sigma_{0y})}{\varepsilon_z} \right) \tag{26c}$$

The term $B$ can be less than -1, typically when $r < 2\sigma_{0,y}$, which can lead to imaginary solutions. This is due to the second order approximation made in converting to cylindrical coordinates. In that case we write a second order approximation based on Taylor series expansion around $B = -1$ as

$$I_{rot}(r,z) = \frac{1}{2\pi r \varepsilon_z \varepsilon_\theta} \left( 1 - \frac{\pi^2(1+B)}{6\varepsilon_\theta{}^2} + \frac{\pi^4(1+B)^2}{40\varepsilon_\theta{}^4} \right) \sum_{i=1}^{i=6} \left\{ exp\left( \frac{-(z-h_i)^2}{2\varepsilon_z{}^2} \right) \right\} \quad for \ B < -1 \tag{27}$$

A similar derivation has been carried out by Green (1980) using different assumptions for the form of Eq. (24).

**3.3 Initial dispersion**

In Sect. 0 and 0 the hourly and annual dispersion parameterizations are described. In both cases initial values for $\sigma_{0(y,z)}$ are required. Since we treat the sources as small area emitters we set the initial $\sigma_{0y}$ to correspond to these areas. A value of $\sigma_{0y} = \Delta y / \sqrt{2\pi} \approx 0.8\,(\Delta y/2)$ will give a maximum sub-grid center concentration equivalent to the concentration that would be found if the emissions were distributed evenly in the sub-grid. We then write the total initial dispersion to be

$$\sigma_{0y} = \sigma_{init,y} + 0.8\frac{\Delta y}{2} \tag{28}$$

In all applications of uEMEP $\Delta x = \Delta y$. The other parameter, $\sigma_{init,y}$, is a specific initial dispersion width for each individual emission source, for example 2 m for traffic. This is generally much smaller than the emissions grids.

The initial value for $\sigma_{0z}$ is also a combination of a specific emission initial dispersion, for example $\sigma_{init,z} = 5$ m for residential wood combustion, but also uses the displacement technique for the plume where the start of the plume is displaced upwind by $\Delta x/2$ allowing the plume to grow vertically over half the sub-grid distance. Tunnel exits are given an initial $\sigma_{init,z} = 6$ m to represent the extended size of the tunnel portals.

**3.4 NO₂ chemistry for hourly means**

The only chemistry included in uEMEP is the $NO_x$, $O_3$ chemical reactions. We use a similar methodology to Benson (1984, 1992) known as the discrete parcel method but use a weighted time scale over which the reactions take place. The following chemical reactions are involved, with $O_x$ ($O_3+NO_2$) and $NO_x$ ($NO+NO_2$) concentrations being conserved:

$$NO + O_3 \rightarrow NO_2 + O_2 \tag{29a}$$

$$NO_2 + h\nu \rightarrow NO + O \tag{29b}$$





$$O_2 + O + M \rightarrow O_3 + M \tag{29c}$$

Eq. (29c) occurs on time scales much faster than the two other reactions and is taken to be instantaneous. The differential equation for the concentration of [NO$_2$] as a function of time is written as

$$\frac{d[NO_2]}{dt} = k_1[NO][O_3] - J[NO_2] \tag{30}$$

where the concentrations are expressed in terms of molecules/cm$^3$ and $J$ is the photolysis rate ($s^{-1}$) for Eq. (29b) taken from the EMEP model (Simpson et al., 2012). The reaction rate $k_1$ for Eq. (29a) is given by

$$k_1 = 1.4 \times 10^{-12} \exp\left(\frac{-1310}{T_{air}}\right) \ (cm^3 s^{-1}) \tag{31}$$

as in the EMEP model and where $T_{air}$ is in the atmospheric temperature (K).

We rewrite Eq. (30) in terms of the dimensionless ratios

$$f_{NO2} = \frac{[NO_2]}{[NO_x]} \ \ and \ \ f_{Ox} = \frac{[O_x]}{[NO_x]} \tag{32a}$$

$$J' = \frac{J}{k_1[NO_x]} \tag{32b}$$

$$t' = tk_1[NO_x] \tag{32c}$$

Eq. (30) then becomes

$$\frac{df_{NO2}}{dt'} = (1 - f_{NO2})(f_{Ox} - f_{NO2}) - J'f_{NO2} \tag{33}$$

The solution to this equation is

$$f_{NO2} = \frac{B}{2}\frac{(1 - A \exp(Bt'))}{(1 + A \exp(Bt'))} + \frac{C}{2} \tag{34}$$


where

$$A = \frac{B + C - 2f_{NO2,0}}{B - C + 2f_{NO2,0}} \tag{35a}$$





$$B = \sqrt{C^2 - 4f_{Ox}}$$ (35b)

$$C = 1 + f_{Ox} + J'$$ (35c)


and $f_{NO2,0}$ is the initial NO$_2$ fraction at time $t'=0$.

This solution is valid for a box model without dilution through dispersion since it does not take into account how changing NO$_x$ and O$_x$ concentrations over the plume travel time will affect the reaction rates. Though this could be accounted for when
applied to a single source with assumed dilution rates, by adding a time dependent diluting term to Eq. (30), this is not practically possible for multiple sources of differing dilutions. The concentrations of NO$_2$ at the start of the plume will be correctly calculated but NO$_2$ concentrations further from the plume will be slightly underestimated, since they do not have the higher initial reaction rates. Eventually the concentrations will reach photo-stationary equilibrium and here too NO$_2$ will be correctly calculated. This special case for photo-stationary equilibrium in Eq. (35) occurs when t' $\rightarrow \infty$ and Eq. (34) becomes


$$f_{NO2} = \frac{C-B}{2}$$ (36)

The non-linear nature of Eq. (34) also means that it cannot be consistently applied to Gaussian models since the shape of the plume will change due to the non-linearity. Despite this, this formulation is more physically realistic than the photo-stationary
assumption often used in local scale air quality modelling or other less physical parameterizations based on empirical fits. See Denby (2011) for an overview of the various NO$_2$ chemistry parameterization methods used with Gaussian modelling.

In order to calculate Eq. (34) in the model application an initial NO$_x$ and O$_x$ concentration must be used and a travel time defined. For multiple sources this travel time will vary so for each calculated sub-grid concentration of NO$_x$ from each
contributing sub-grid source ($n_s$ sources) a travel time, $t_s$, is calculated based on the distance and wind speed. This is weighted based on the contribution of each source to the total sub-grid NO$_x$ concentration. This provides a final weighted travel time $t_w$ that is applied in Eq. (34). This ensures that nearest of the contributing sub-grids, often with the highest contributing NO$_x$ concentrations, are given a higher travel time weight. A minimum distance, and hence time, of half a sub-grid is applied when calculating travel times.


$$t_w = \frac{\sum_{s=1}^{n_s} t_s [NO_x]_s}{\sum_{s=1}^{n_s} [NO_x]_s}$$ (37)

### 3.5 NO$_2$ – NO$_x$ conversion for annual means

When annual mean data are used then the hourly mean formulation cannot be applied. Instead we use an empirically based conversion of NO$_x$ to NO$_2$ based on the type of formulation from Romberg (1996) and updated by Bächlin and Bösinger





(2008). 3 years of Norwegian NO₂ measurements, 82 measurements in all, have been used to determine this relationship, Fig. 5.

$$[NO_2] = \frac{a\,[NO_x]}{[NO_x]+b} + c\,[NO_x] \qquad (38)$$

The fitted constants are determined to be a=20, b=30 and c=0.23. The estimated uncertainty in this conversion is around 10%, based on the normalized root mean square error of the fitted and observed NO₂ concentrations.

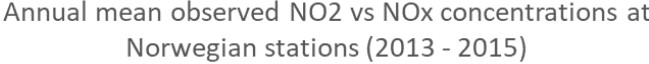

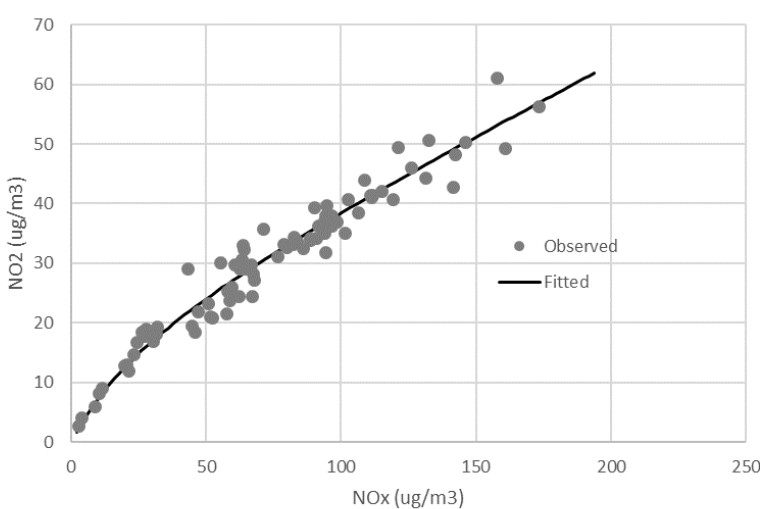

**Figure 5. NO₂ verses NOₓ annual mean concentrations for all stations in Norway in the period 2013-2015. The fitted curve is based on Eq. (38).**


### 3.6 Implementation

### 3.6.1 Sub-grid domains

Within uEMEP individual domains are defined with differing resolutions and sizes, dependent on which modelling parameter is represented. Separate domains and sub-grid resolutions are defined for each of the emission sources, for the time profiles of
each emission sources, for the meteorological data, for the population data and for the receptor sub-grid concentrations. None of these are required to have the same resolution or size, however, the highest resolution emission sub-grid will define the receptor sub-grid resolution, since there is no need to calculate on higher resolution sub-grids than is provided by the emissions. For emission sub-grids with lower resolution than the final receptor sub-grid domain then the dispersion calculations are first





carried out at the same resolution as the emission domain and then bilinearly interpolated to the receptor sub-grid. For most
urban applications this means that the choice of traffic sub-grid resolution defines the highest resolution sub-grid.

Emission sub-grids also contain properties for the dispersion calculations, such as initial dispersion parameters and emission
heights. Each emission sub-grid has only one emission height $h_{emis}$, one $\sigma_{init,z0}$ and one $\sigma_{init,y0}$ for each emission source type.
When multiple emissions from the same source type are placed in an emission sub-grid then the emission parameters are
weighted by each individual emission. This is most relevant for industrial emissions which may have different emission heights
from separate stacks within a single emission sub-grid.

### 3.6.2 Selective sub-grid calculations

uEMEP does not necessarily calculate concentrations at all receptor sub-grids. Only sub-grids which are within $3\sigma_y$ of a plume
centre line will be calculated and also downwind selection is used (Supplementary material, Sect. S3.4.2). In addition, a number
of selections can be made allowing quicker calculations for particular applications. These include:

1. Calculation at defined receptor points, usually corresponding to measurement stations. In this case uEMEP calculates
   the surrounding 9 sub-grids and uses bilinear interpolation to extract the concentrations at the required receptor
   position.
2. Calculation at population grids. In this case concentrations will only be calculated at grids with non-zero population.
This provides quicker exposure calculations than if the entire region was calculated
3. High density calculations near sources. A routine for selecting a higher density of sub-grids near sources may also be
   used to speed up calculations. This applies most often to traffic emission sub-grids that are near surface and with large
   gradients near the source. This is less useful for higher release sources as their maximum impact occurs further
   downwind than their emissions. After calculation the lower density receptor sub-grids are interpolated into the rest of
the receptor sub-grids, providing a full receptor sub-grid domain

### 3.6.3 Model inputs and outputs

Input data comes from a variety of sources and the formatting of these sources varies. Emission input data is generally in text
format whilst meteorological files are read from netcdf files.

Output of the model is in the form of netcdf files for either gridded data or point data, if receptor points have been defined. In
both these files output includes the total concentrations of the pollutants along with the source contribution from each of the
emission sources used in the calculation. Speciation of PM from EMEP can also be included in the output files, along with
emissions, meteorology and population data.





## 4. Implementation in the Norwegian air quality forecast and analysis system

Though uEMEP has been applied in a number of applications we select the Norwegian forecast and analysis system
(Norwegian air quality forecasting service, 2020) as an example. This application started operationally in the winter of 2018-
2019 and provides daily forecasts of air quality for all of Norway two days in advance at sub-grid resolutions of between 250
and 50 m. In addition, the same system is used to calculate air quality retrospectively for analysis and planning applications
(Norwegian air quality expert user service, 2020). The compounds $PM_{2.5}$, $PM_{10}$, $NO_2$, $NO_x$ and $O_3$ are calculated. For each of

545     these the local source contributions are determined separately for traffic exhaust, traffic non-exhaust, residential wood
combustion, shipping and industry. A cascade of models are used starting with EMEP MSC-W at 0.1° European domain,
EMEP MSC-W at 2.5 km Scandinavian domain and uEMEP 250-50 m Norway only, Fig. 6.

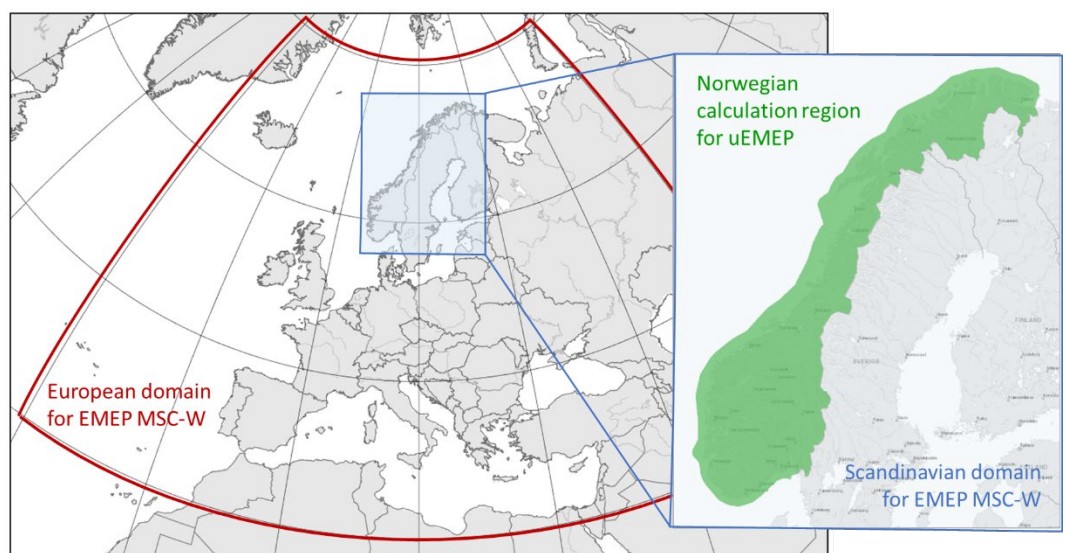

550
**Figure 6. Model domain for the European and Scandinavian EMEP MSC-W calculations and the uEMEP calculations
(©kartverket/norgeskart.no).**

### 4.1 Calculation steps

555     We describe below the implementation steps used in the Norwegian forecasting and analysis system. This implementation of
uEMEP uses the independent emission and replacement downscaling method (method 2 in Sect. 2). The following steps are
carried out:

1.    High resolution emission data for Norway are calculated for each forecast (Sect. 4.2) and are aggregated into the
EMEP MSC-W Scandinavian model grid. Some of these emissions require meteorological data.



2. The EMEP MSC-W model is used to calculate the large scale concentration distribution on an hourly basis, nesting from a European domain (~0.1°) to the Scandinavian domain (2.5 km), Fig. 6. Within Norway the aggregated high resolution aggregated emissions are implemented. Both EMEP calculations provide the local fraction (Sect. 2.3) in a region of 5 x 5 EMEP grids.

The following three steps are then undertaken to calculate the uEMEP concentrations:

3. For the Norwegian forecast system the entire country is split into 1864 separate tiles of varying sizes and resolutions; the resolution depending on the population and emission sources within each tile. Tiles with resolutions of 250 m can be as large as 40 x 40 km$^2$ whilst tiles with resolutions of 50 m are no larger than 5 x 5 km$^2$. Tiling the calculations is a form of external parallelisation and is optimised for both runtime and memory use. A two day forecast run on 196 processors takes roughly one hour of CPU time.

4. The high resolution emission data from the various source sectors (Sect. 4.2) is placed into the emission sub-grids in uEMEP. These are between 50 – 250 m in width, depending on the emissions available and on the population density of the region. Emission grid domains extend beyond the size of each tile so that the calculations are consistent over tile borders.

5. uEMEP Gaussian dispersion modelling is applied (Sect. 0) using the sub-grid emissions as sources and the concentrations are calculated at each sub-grid. Only sub-grid emissions within a region defined by a 4 x 4 EMEP grid area are included in the sub-grid calculation, i.e. 10 x 10 km$^2$, corresponding to the extent of the moving window. This 4 x 4 limit guarantees that the calculation will always be carried out within the EMEP 5 x 5 local fraction region.

The final steps combine the EMEP gridded concentrations with the uEMEP sub-grid concentrations in the following way:

6. At each sub-grid the non-local contribution from the neighbouring 4 x 4 EMEP grids is calculated, Sect. 2.4. The calculation is carried out for each source sector and each primary compound

7. The uEMEP calculations are then added to the non-local EMEP concentrations. In the case of PM then all non-primary species are also added to the local and non-local EMEP primary concentrations

8. For NO$_2$ the chemistry (Sect. 3.4) is applied to determine NO$_2$ and ozone for each sub-grid

9. Sub-grid concentrations and their contributions are saved along with the PM speciation from EMEP in netcdf format.

10. The forecasts are made available to a public website through an API and Web Map Tile Server (Norwegian air quality forecasting service, 2020)

The system is schematically illustrated in Fig. 7. The following sections describe some steps in more detail.



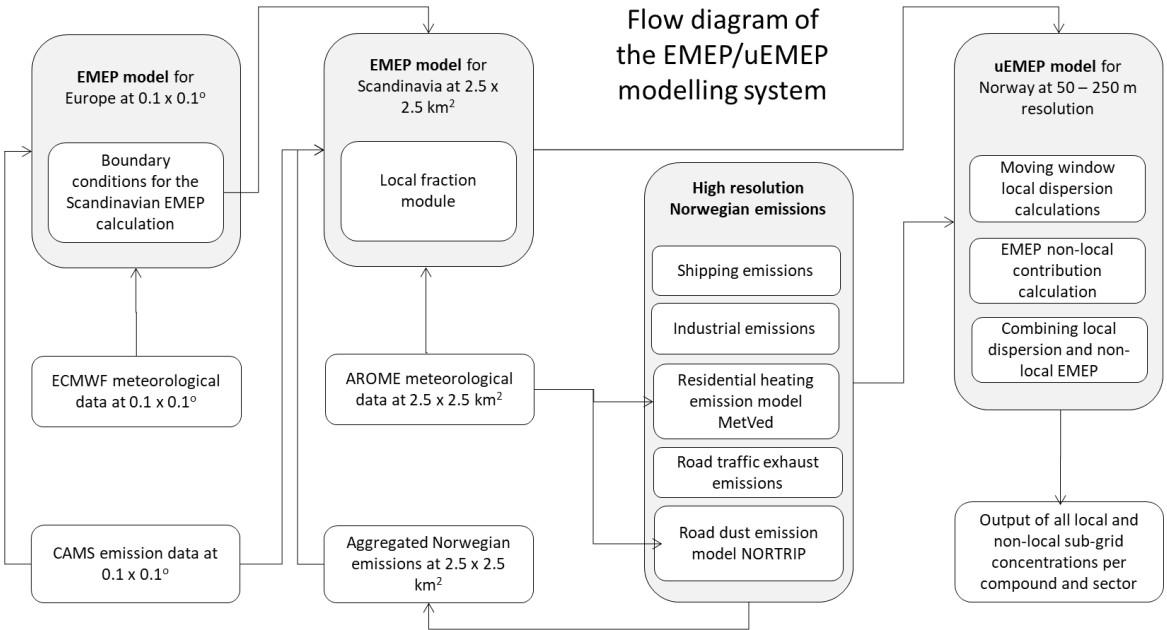

**Figure 7. Flow diagram showing the various components of the Norwegian EMEP/uEMEP forecast system.**

## 4.2 Emissions

The EMEP calculations make use of the CAMS-REG-AP_v1.1 regional anthropogenic emission dataset everywhere in Europe (Kuenen et al., 2014; Granier et al., 2019). Only in the 2.5 km Scandinavian calculation, and only in Norway, are the emissions replaced with the aggregated high resolution dataset. The alternative emissions used in the calculations for Norway are:

- Road traffic exhaust emissions
- Road traffic non-exhaust emissions
- Residential wood combustion
- Shipping
- Industry

These emission sources are described in the supplementary material, Sect. S4.2. For other sectors the CAMS-REG-AP_v1.1 emissions are also used in Norway, but these emissions are not downscaled using uEMEP.

## 4.3 Meteorology

The meteorological forecast data used for the European EMEP model calculations is based on the Integrated Forecasting System (IFS, 2020) from the European Centre for Medium-Range Weather Forecasts (ECMWF, 2020). The Scandinavian EMEP model calculation uses the AROME-MetCoOp model for modelling meteorology over Scandinavia (Müller et al.,



2017). This last model calculates meteorology at a resolution of 2.5 km and provides forecasts for 66 hours in advance. The
EMEP MSC-W Scandinavian domain uses the same gridding and projection as the meteorological forecast model but in a
smaller domain.

### 4.4 EMEP MSC-W model implementation

The European EMEP MSC-W model calculation is based on the same daily forecast provided for the Copernicus Atmosphere
Monitoring Service (CAMS, 2020; Tarrason, 2018) but is run independently and provides boundary conditions for the
Scandinavian implementation of EMEP MSC-W at 2.5 km. The Scandinavian EMEP MSC-W calculation includes the
Norwegian emission sources described in Sect. 4.2 and also delivers the necessary local fraction information for use in uEMEP.

### 4.5 uEMEP model implementation

uEMEP calculates concentrations for all of Norway on grids with resolutions between $50 - 250$ m using 1864 individual tiles
as described in Sect. 4.1. The resolution of these tiles is defined by the population density and road density information.  Tiles
with higher population density use 50 m resolution, whilst tiles with lower population density but some traffic have a resolution
of 125 m. Tiles with very low traffic density but with shipping or wood burning emissions have a resolution of 250 m,
corresponding to the emission resolution. Separate calculations are carried out at measurement sites, 72, with a sub-grid
resolution of 25 m. An example of a $PM_{10}$ forecast is shown in Fig. 8.

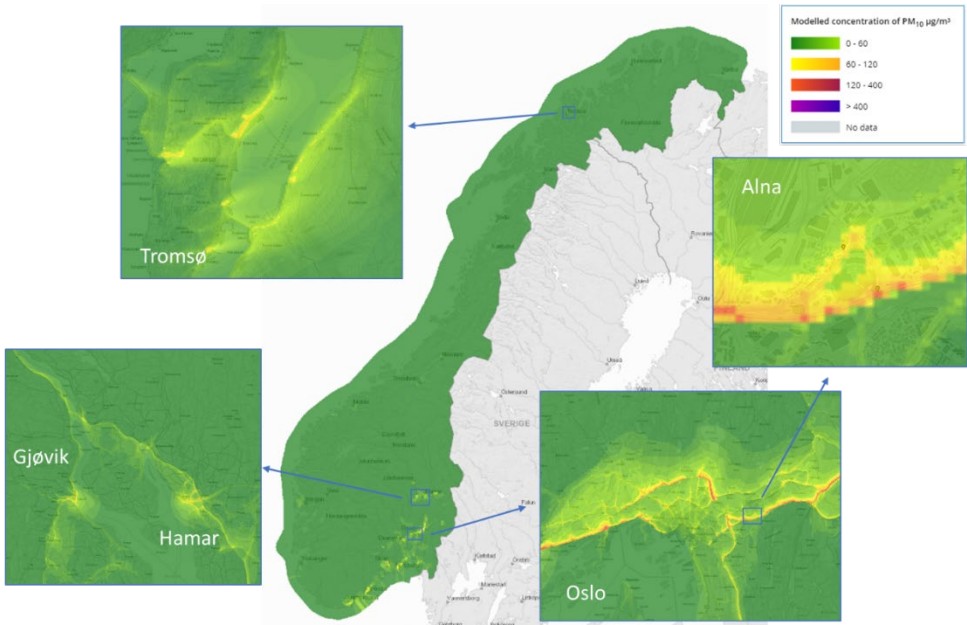


**Figure 8: Example maps of $PM_{10}$ concentrations taken from the forecast 24.02.2020 18:00 UTC. Resolution in populated city regions
is 50 m. High $PM_{10}$ concentrations along roads are mainly the result of road dust emissions (©kartverket/norgeskart.no).**



# 5. Results

## 5.1 Validation against observations for the Norwegian forecasting and assessment system

In the supplementary material we provide a complete and detailed statistical validation for the year 2017. Here we present a visual summary of results for $NO_2$, $PM_{10}$ and $PM_{2.5}$ for the same year. In 2017 there were 72 operational air quality stations. Not all stations measure all components so the total number of available stations for $NO_2$ and PM with coverage of more than 75% is between 34 – 45. The station positions are shown in Fig. 9.

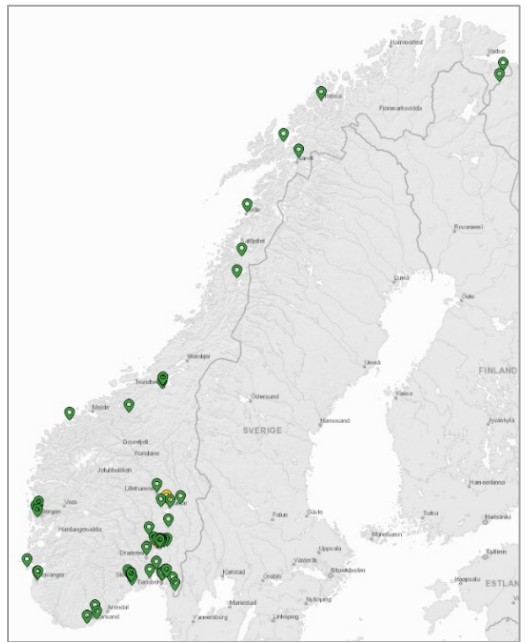

**Figure 9. Positions of all 72 monitoring stations in Norway. 33 for $PM_{2.5}$, 36 for $NO_2$, 45 for $PM_{10}$, 8 for $O_3$ (©kartverket/norgeskart.no).**

### 5.1.1 NO₂

Fig. 10 shows the comparison of modelled and observed $NO_2$ for annual average at each station (scatter plot) and daily mean temporal profile averaged over all stations. Included in the scatter plot are the Scandinavian EMEP MSC-W results at 2.5 km.

The spatial correlation is quite high, $r^2=0.80$ for uEMEP with little negative bias (FB=-6.7%). The temporal variation over the whole year is also well represented when averaged over all stations ($r^2=0.79$).





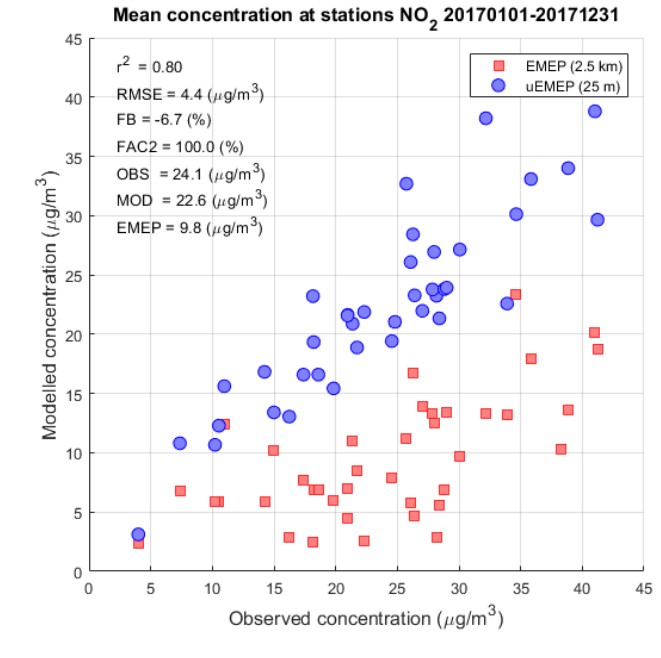

(a)

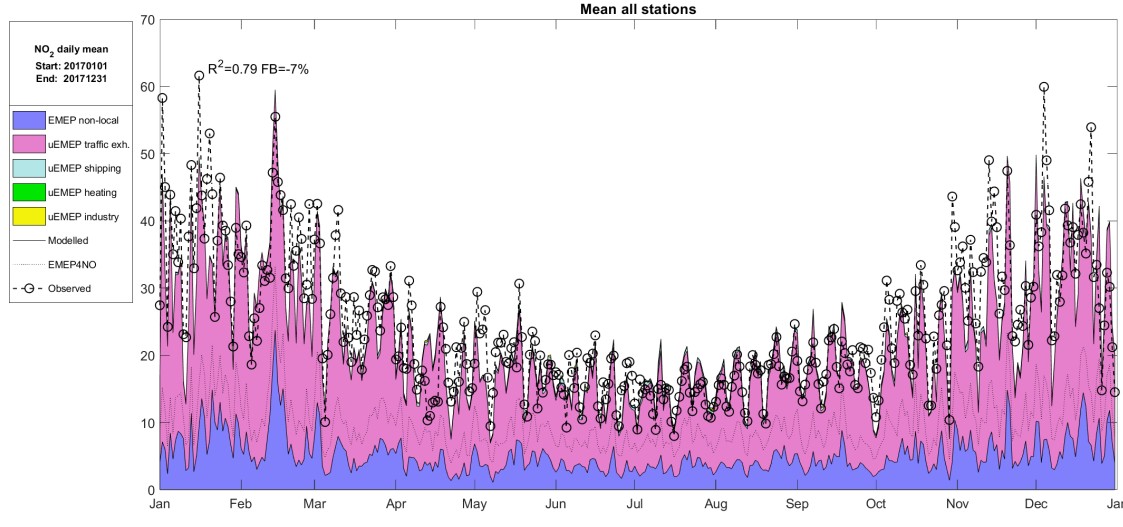

(b)

**Figure 10. Scatter plot comparison of modelled and observed NO₂ for annual average at each station (a) and daily mean temporal profile averaged over all stations (b). Source contributions are shown for the temporal modelling results along with the EMEP 2.5 km calculation (EMEP4NO). 36 stations are used in the comparison.**





**5.1.2 PM$_{10}$**

Fig. 11 shows the comparison of modelled and observed PM$_{10}$ for annual average at each station (scatter plot) and daily mean temporal profile averaged over all stations. Included in the scatter plot are the Scandinavian EMEP results at 2.5 km. The spatial correlation is low, $r^2$=0.29 for uEMEP with little negative bias (FB=-11.2%). The temporal variation over the whole year is well represented when averaged over all stations ($r^2$=0.61) but the model has a negative bias of 4 µg/m$^3$ over much of the summer period. Road dust events in the spring time are well captured by the emission model NORTRIP.


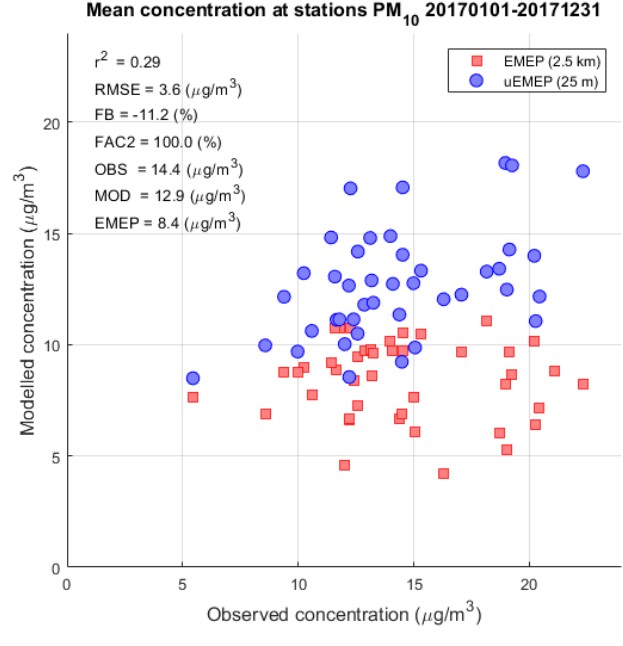

(a)





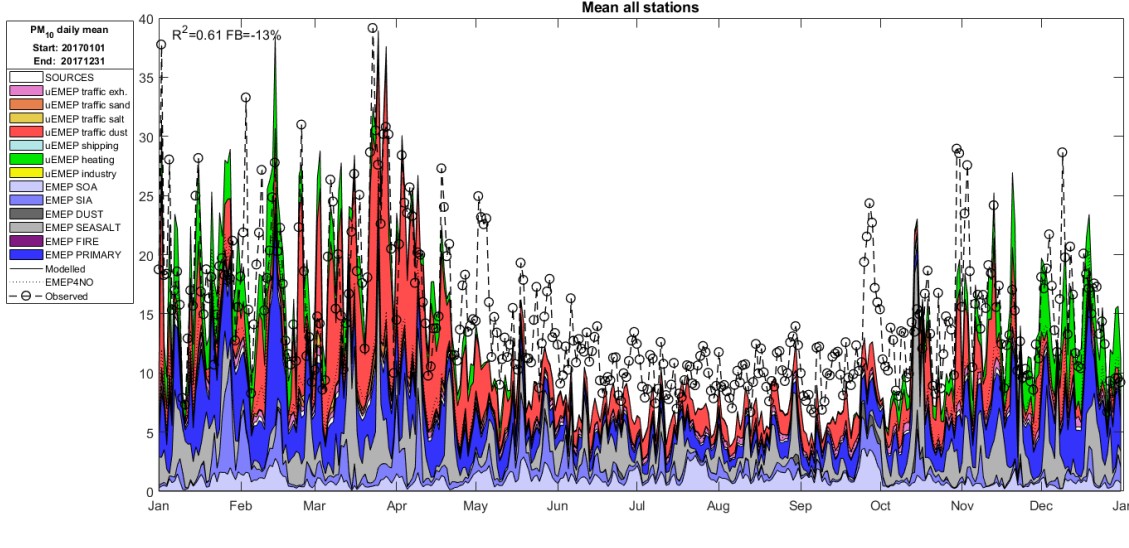

(b)


**Figure 11. Scatter plot comparison of modelled and observed PM₁₀ for annual average at each station (a) and daily mean temporal profile averaged over all stations (b). Source contributions from both uEMEP and EMEP models are shown for the temporal modelling results along with the EMEP 2.5 km calculation (EMEP4NO). 45 stations are used in the comparison.**

### 5.1.3 PM₂.₅

Fig. 12 shows the comparison of modelled and observed PM$_{2.5}$ for annual average at each station (scatter plot) and daily mean temporal profile averaged over all stations. Included in the scatter plot are the Scandinavian EMEP results at 2.5 km. The spatial correlation is good, r$^2$=0.49 for uEMEP with little negative bias (FB=-12.5%). The temporal variation over the whole year is well represented when averaged over all stations (r$^2$=0.67) but the model has a negative bias of 2 µg/m$^3$ over much of the summer period. Residential wood combustion (heating) events in the winter are well captured by the emission model

MetVed.



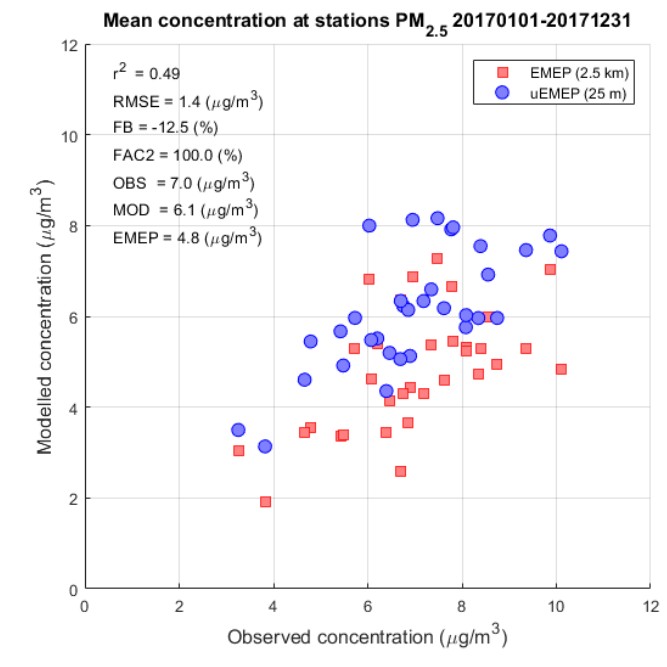

(a)

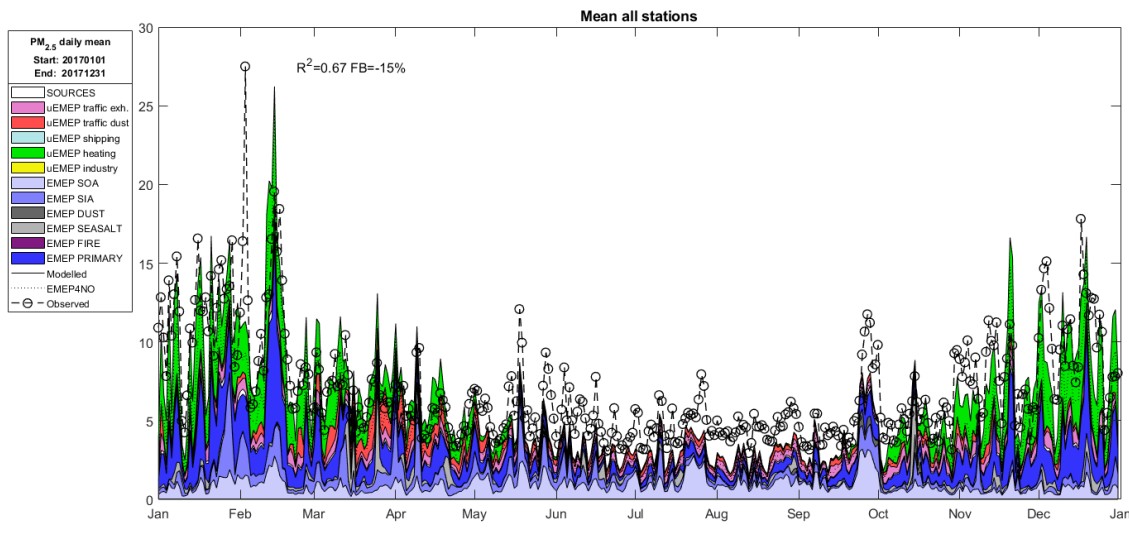

(b)

**Figure 12. Scatter plot comparison of modelled and observed PM$_{2.5}$ for annual average at each station (a) and daily mean temporal profile averaged over all stations (b). Source contributions are shown from both uEMEP and EMEP models for the temporal modelling results along with the EMEP 2.5 km calculation (EMEP4NO). 33 stations are used in the comparison.**



## 5.2 Verification and sensitivity tests

In addition to the validation against monitoring data a number of verification and sensitivity experiments have been undertaken with the model. These include:

- Comparison of single annual mean calculations with the mean of hourly calculations
- Sensitivity to the moving window size
- Sensitivity to the choice of resolution

- Sensitivity to the temperature dependence of $NO_x$ exhaust emissions
- Sensitivity to the choice $NO_2/NO_x$ initial exhaust ratio

These sensitivity tests are provided in the Supplementary material (Sect. S5). We present only conclusions from these.

### 5.2.1 Comparison of single annual mean calculations with the mean of hourly calculations

In Sect. 3 we describe two methods for calculating dispersion. One is based on the hourly meteorological and emission data, Sect. 3.1, and the other on annual mean data, Sect. 3.2. A rotationally symmetric dispersion kernel, Eq. (25), is used for dispersion of the annual mean emissions. Tests using the same dispersion parameters in both annual and hourly calculations, Sect. S5.1, give very similar results for both methodologies indicating the validity of the annual mean calculation. When $K_z$ based dispersion, Eq. (15-23), is used in the hourly calculations then there is a larger difference between the two methods

because of the difference between the two dispersion parameterisations. We conclude that the time saving advantage of the single annual mean calculation, approximately 10000 times faster, and the similarity to the hourly calculation make it an efficient and valid method for calculating high resolution annual maps of air quality.

### 5.2.2 Sensitivity to the moving window size

The size of the moving window region within which uEMEP calculates local high resolution concentrations should impact on

the results since smaller moving windows will include less locally modelled contributions and more non-local EMEP contributions. This has been verified in a sensitivity study, Supplementary material Sect. S5.2. In this sensitivity experiment the moving window size was varied from $n_{mw}$ = 1 to 8 EMEP grids and calculations were made at existing measurement sites. The mean concentrations are shown to be quite insensitive to the choice of this region, particularly for $PM_{10}$. Generally the reduction in the local contribution is well balanced with the increase in the non-local contribution when reducing the size of

the moving window, verifying the methodology. It is recommended to use a minimum of 2 EMEP grids for the moving window region.





### 5.2.3 Sensitivity to the choice of resolution

The choice of sub-grid resolution will impact on the calculated concentrations, both in concentration levels and in spatial distribution. An experiment where a range of sub-grid resolutions were tested, from 15 m to 250 m, was carried out, Sect. S5.3.

Calculations were made at the positions of the Norwegian monitoring sites, most of which are traffic sites. The results showed that even at resolutions of 250 m the mean concentrations for all stations were very similar. At 100 m resolution, compared to the reference of 25 m, the difference in annual mean was no larger than 15% at any one station with a normalised root mean square error (NMRSE) of 6%. The NRMSE increased to 11% for the 250 m calculation with a maximum deviation of 40% at one station. We conclude that 100 m resolution will provide good concentration estimates for near road calculations though

higher resolutions may be required, depending on the application.

### 5.2.4 Sensitivity to the temperature dependence of NOx exhaust emissions

The temperature dependence of the $NO_x$ traffic exhaust emissions was assessed by running the model with and without this dependency, Sect. S5.4. With this correction the results show a significant improvement in the station mean time series correlation (from $r^2$=0.68 to 0.80) and improved correlation in both the daily (from $r^2$=0.56 to 0.60) and annual (from $r^2$=0.76

to 0.78) mean calculations. Bias is also reduced from -20% to -3%. The correction factor used, Eq. (S13), still requires further evaluation and should be considered only as an initial estimate.

### 5.2.5 Sensitivity to the choice NO₂/NOₓ initial exhaust ratio

In the calculations shown in Fig. 10 for $NO_2$ an initial $NO_2/NO_x$ exhaust emission ratio of 0.25 was used. This reflects the large portion of diesel vehicles used in Norway and the high $NO_2/NO_x$ ratio of these (Hagman, 2011). However, comparison of

modelled ratios of $NO_2/NO_x$ indicate this ratio may be too high. This was assessed by running the model with three different ratios, 0.15, 0.25 and 0.35. The results, Sect. S5.5, show that an $NO_2/NO_x$ ratio of 0.15 most closely fits the observed ratio and this ratio will be implemented in further applications of the model.

## 6. Discussion

The aim of uEMEP is to provide downscaling capabilities for the EMEP MSC-W model with the intention of improving

exposure estimates and more realistic concentrations at high resolution over large areas. The example application provided, the Norwegian air quality forecast and expert user service, is an example of how high resolution coverage over large regions (countries) can be achieved. The validation carried out in Sect. 5.1 shows that the modelling system provides moderate to good comparison with observations. The best results are for $NO_2$, chiefly because we have the best information concerning emissions that contribute to these concentrations, i.e. traffic exhaust. The lower correlation of PM is indicative of the difficulties in

modelling emissions such as residential wood burning and road dust emissions. That $NO_2$ is well modelled indicates that the



problems lie largely with the emissions, rather than the dispersion model itself. In addition a large proportion of PM is due to medium to long range transport and secondary formation of particles. This is not part of the uEMEP model but relies on the EMEP MSC-W model and the emissions included there.

The strength of the modelling system is in the integration of uEMEP with EMEP through the use of the local fraction. This allows downscaling anywhere within an EMEP domain provided that suitable proxy data is available for the downscaling. This is an important aspect of the modelling and is the link that can bind the regional and local scale emission communities. Usually the proxies used for regional scale emission inventories are not available to the user so that exactly how these emissions are made, quantitatively, is unknown to the user. In addition, as the resolution of regional scale emission inventories increase so
too does the need for improved spatial distribution proxies. Population density, successfully used to distribute a range of emission sectors on low resolution grids ( > 10 km) is no longer suitable for many sectors since at high resolution the emissions are no longer correlated with population. This was discussed in an earlier paper, Denby et al. 2011, and remains problematic.

When implementing uEMEP it is highly desirable that the emissions used in both uEMEP and EMEP models are consistent
with one another. This has been achieved for the Norwegian application for the sectors traffic, domestic heating, shipping and industry. However other sources, such as other mobile combustion sources associated with construction and other activities, are not included. These can be of importance locally even if they are not significant on the regional scale. There is no clear methodology available on how to implement these emissions at the required resolution.

The modelling system has limitations. Currently only primary emissions, with the exception of $NO_2$ formation, are dealt with. Some secondary formation of particles will likely occur within the local region used for the uEMEP model domain but these are not currently accounted for. uEMEP is also a Gaussian model that does not take into account obstacles of any type. When achieving resolutions of 50 m then buildings start to play an important role in the transport and dispersion. The region covered by the local scale modelling, the moving window region, is necessarily limited in extent. Sensitivity studies show that this has
limited impact on mean concentrations but for source sectors such as industry, that are released at height, the limited calculation region may not be large enough to include all of the plumes impact region.

In many ways the increase in resolution to almost street level puts new demands on the modelling system that were not necessary to consider previously. For regional scale modellers the downscaling can provide considerable improvement to
regional calculations. However, from a local scale modelling perspective, the local scale information may not be of sufficient quality to be useful to local users. This is most important when only proxy data is available for downscaling rather than actual bottom up emissions. In the end, if high resolution modelling is to be used at the local scale then similarly high quality emission data will be required if the results are to be useful to users.




There are a number of aspects of the modelling system that can be, and are being, improved. These include:

- Implementation of dry and wet deposition in uEMEP, currently not included in this version
- Improving the annual mean dispersion kernel dispersion parameters to be more consistent with the hourly $Kz$ methodology
- Implementing necessary secondary formation of PM in uEMEP
- Further assessment of the $K_z$ Gaussian plume methodology
- Refinement of the temperature dependence of $NO_x$ traffic exhaust emissions

A number of aspects were not treated in this paper but will be topics of further studies. These include population exposure and the impact of resolution, trend assessment in emissions and analysis of road dust emissions for all of Norway. In addition, the modelling system is being applied in a number of different countries and results of these applications will be further described and assessed.

## 7. Conclusion

This paper presents and documents a new downscaling model and method (uEMEP) for use in combination with the EMEP MSC-W chemical transport model. Process descriptions and parameterisations within the uEMEP model are provided and the methodology for combining uEMEP with EMEP MSC-W local fraction calculations is elaborated. An example application, The Norwegian air quality forecast system, is presented and validation for the year 2017 at all available Norwegian air quality stations is provided. A number of verification and sensitivity studies are summarised in the paper and expanded in the Supplementary material.

The uEMEP model provides a new methodology for downscaling regional scale chemical transport models but can currently only be applied together with the EMEP MSC-W model since this is the only model with the necessary local fraction calculation. The uEMEP model is based on Gaussian modelling that has existed for many years but it does use specific parameterisations to describe the dispersion parameters in order to be compatible with the EMEP model application.

uEMEP can provide improved exposure estimates if suitable proxy data for emissions are available and can be applied to regions as large as the regional scale CTM in which it is imbedded. It can also represent concentrations down to street level and is comparable with traffic monitoring sites. This makes it a unique system for assessment, policy application and forecasting purposes.



# Code and data availability

The current version of uEMEP is available from Github (https://github.com/metno/uEMEP) under the licence GNU Lesser General Public License v3.0. The code is written in fortran 90 and is compilable with intel fortran (ifort). The code does not support gfort as a compiler at this time. The exact version of the model used to produce the results used in this paper is archived on Zenodo (DOI: 10.5281/zenodo.3756008), as are input data and scripts to run the model and produce the plots for all the simulations presented in this paper (DOI: 10.5281/zenodo.3755573).

# Author contribution


Bruce Rolstad Denby is the lead author of the article and developer of the uEMEP model. Michael Gauss and Hilde Fagerli internally reviewed and contributed to the writing of the article. Peter Wind is the developer of the local fraction methodology in the EMEP MSC-W model and provided text on this. Qing Mu carried out the EMEP MSC-W calculations for this article and contributed to the text. Eivind Grøtting Wærsted carried out the validation of the uEMEP model and contributed to the 810 text. Alvaro Valdebenito and Heiko Klein provided the technical support for carrying out the modelling and contributed to the uEMEP code and associated scripts for its implementation.

# Acknowledgements

uEMEP was developed and applied with financing from the Research Council of Norway (NFR) project 'AirQuip' (grant no. 267734), The Norwegian Public Roads Administration (Statens Vegvesen), the Norwegian Environment Agency 815 (Miljødirektoratet) and the Ministry of Climate and Environment (Klima- og miljødepartementet).

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
