# Peer review of "Description of the uEMEP\_v5 downscaling approach for the EMEP MSC-W chemistry transport model"

_Geoscientific Model Development, 2020_

## Referee Comment (RC1) · Anonymous Referee #1 · 22 Jul 2020

General comments:

This paper presents a novel methodology to downscale a chemical transport model (CTM) result to local scales (i.e., tens of meter resolution) over large geographical domains without double counting of emissions. A Gaussian dispersion kernel is implemented in the uEMEP model to estimate the local contribution of emission sources at fine resolutions while still considering the contribution of the EMEP CTM. The new capability of the EMEP model to quantify the local and non-local contribution of neighbouring emission sources to air pollutant concentrations in a specific grid cell of the model is used. A refined combination of the non-local contribution of the EMEP concentrations with the result of the Gaussian model within uEMEP provides significant improvements in places near emission sources like traffic sites. The new uEMEP provides a flexible treatment to refine the results of the CTM as desired over large domains, with still some limitations in street-canyon environments. The manuscript describes in detail the downscaling methodology and the Gaussian model implemented in uEMEP. Some parts of Section 2 describing the coupling of the models are difficult to follow and some clarification would help the reader. The work is well presented and should be accepted for publication in Geoscientific Model Development after minor revisions.

Specific comments:

- Line 46: the Authors could mention the OSPM model as an example of a street-canyon model to complement the overview of local scale models.

- Line 95: clarify if this option implies that emissions in the EMEP grid cell are not consistent with the ones in the sub-grid cells.

- Line 101: some discussion on the implications of using such inconsistent chemistry treatments in uEMEP and EMEP would be appreciated. As uEMEP is intended for applications over wide regions with significantly different chemical regimes, the simple chemistry may perform better in some environments than others.

- Line 116 and 140: the term Csg_nonlocal(i,j) is the more complex to understand. Perhaps, an equation describing how is computed would help the reader. I appreciate the effort of the Authors to explain the method with Figure 1 and 2 and Section 2.3, but it is still confusing how the local and non-local contributions of the EMEP grids are used in the computation of the Csg_nonlocal term.

- Line 147: More details on Wind et al. (2020) methodology would be appreciated in the manuscript. Considering that the local fraction estimate links emissions with concentrations, the Authors could clarify how the chemistry is treated once the tagged emissions are dispersed in the EMEP grid cells. Are tagged primary pollutants emitted

as inert tracers or limited chemistry is considered? The details are provided in Wind et al. (2020), but the reader would appreciate some further descriptions of the method and limitations in the present manuscript.

- Line 152: Provide which fraction of the total contribution is missed in the local fraction estimate when using few EMEP grid cells.

- Line178 Eq. 6: Why this is not divided by the sum of the weights? Following the example in Fig. 1, you use more than 9 EMEP grid cells (adding their concentration) to obtain the local contribution of the moving window over the i,j sub-grid cell. This results with a local contribution overestimated somehow if nmw>1. For the case nmw=1, the expression seems good as the sum of the weights would be 1.

- Line 233 Eq. 10: Why $C_g(i,j,s)$ is divided by nsource if it is already the concentration of a specific source?

- Line 295: I suggest introducing in this section the meandering and traffic term described in the supplementary material. Some variables in the equations are not defined just before or after presenting the equation. It would help the reader to introduce all the terms after the equations and specify which ones will be further described in subsequent sections.

- Line 330: Mention the floor value of the wind speed imposed in the model in this part of the manuscript. Some details are only presented in the supplementary material.

- Line 414: a table with the sigma_init_y values per emission source would be appreciated.

- Line 518: an order of magnitude of the maximum distance allowed in the dispersion of the Gaussian model would be appreciated (i.e., 250 m).

- Line 583: Is ozone also a product used from uEMEP? Is there any evaluation done for this pollutant?

- Line 631: Some discussion about the improvement in the daily cycle of the uEMEP results compared with EMEP would be appreciated. Local models use to improve the traffic peaks but also may inherit issues with the temporal profiles and the boundary layer evolution. The validation section could be improved introducing some discrimination between types of sites (rural, industrial, suburban, urban). I suggest presenting all the material of subsections 5.1.1, 5.1.2 and 5.1.3 under section 5.1 as those sections consist only in a single paragraph.

- Line 653: What missing processes could explain the remaining bias during the summer period in both PM10 and PM2.5?

- Line 700: it is counter-intuitive having more non-local EMEP contributions with smaller moving windows. Could the Authors clarify this in the text? If less EMEP grid cells are used in the moving window, less non-local contributions would be expected.

- Line 796: There are still some street-canyon processes that uEMEP cannot represent, particularly in compact cities with high street aspect ratios. The Authors should mention this in this last concluding remark.

Technical comments:

- Line 29: the acronym CTM is used several times in the manuscript but defined in Line 71. Please, define the acronym already in the introduction and use directly the acronym in the rest of the manuscript.

- Line 51: use coma instead of a semi-colon in the reference

- Line 58: the reference Wind et al. (2020) is not provided in the reference section.

- Line 154: fix the Section number. Here and in other parts of the manuscript, the number of the reference to specific sections is 0.

- Line 245: Use Eq. instead of Equ. in the Figure caption.

- Line 362: Monin–Obukhov is mistyped in different parts of the manuscript.

- Line 362: the Monin-Obukhov length and the surface roughness have already been used before in the manuscript. Define them there only once.

- Line 371 Table1: please, use consistent notation for the boundary layer height and Monin-Obukhov length. Both have been introduced before as H and L.

- Line 407 and 574: fix the section number that appears in the reference Sect. 0.

- Line 646: the statistics presented in panel (a) should be introduced in the caption specifying for which model are computed. In panel (b), the Authors could remove the shipping and industry labels in the legend as no information is shown in the figure.

- Line 661: There is too much information in Figure 11. I suggest presenting the non-local contribution of EMEP and not the detailed composition of it. Though of interest, it is impossible to appreciate EMEP4NO line and some artefacts appear as the white contribution above EMEP PRIMARY blue fraction.

- Line 679: avoid using subsections that consist of a single paragraph.

- Line 719: to be consistent with the supplementary material the coefficient of determination of the station mean time series of uEMEP should be 0.79, not 0.80. Harmonise the number in both documents.

- Line 728: I suggest merging Sections 6 and 7.

Comments Supplementary material:

- Line 13: Use section S1 instead of S3 and number accordingly the rest.

- Line 106: It should be Eq. (15a).

- Line 242: Why the inverse of the wind speed is used instead of wind speed?

- Line 314: In the figure caption, it should be Fig. S4 instead of S2.

- Line 328: The observation measurement could be provided in Fig. S6.

[Figure]

- Line 368: Why Figure S8a is different from Figure 10b? The caption describes the same results.

- Line 385: A value of 0.1 would likely provide an even closer fit to observations.
* * *

---

## Referee Comment (RC2) · Anonymous Referee #2 · 6 Sep 2020

This is an interesting and extensive description of the uEMEP downscaling scheme for the EMEP MSC-W CTM. The uEMEP scheme represents a major step in downscaling of ambient NOx/NO2 and PM concentrations to local levels and is well documented in this manuscript, alongside with validation and sensitivity analysis. Though the text is not always easy to read, it is in general well written. I have only a couple of minor questions and issues that should be clarified in the text, which should help to improve readability in some parts.

Specific comments

- line 33 / "near street level modelling": What is then the ambition of the model? Is

it supposed to represent concentrations at roadside monitoring sites or background sites?

- What is the meaning of resolution <100m when there is no local topography modelling involved? Wouldn't building layout, air flows in the street canyon etc need to be accounted for at these very local scales?

- line 85: Which source sectors are included in the uEMEP downscaling calculations? Traffic, residential, any other? Should be mentioned somewhere in Sect 2.1

- line 150. "neighbour cells" sounds as if only +/- 1 in each direction but I understand from the next sentence that the local fraction region can be quite large. Please clarify in the text.

- line 153. Perhaps I missed it but it would be good to have a paragraph somewhere that explains the difference between the different domains (uEMEP vs local fraction vs moving window) as it is a bit confusing to the reader.

- Sect 2.3-2.4: These sections are difficult to follow, I would suggest restructuring 2.3 and 2.4 into one (The second sentence of 2.3 already refers to 2.4).

- Sect 2.4: This is rather complicated to follow for an effect that is probably second-order. How much is gained by the complicated moving window calculation of non-local contributions at sub-grid resolution? With a reasonably big local fraction tracking domain, the difference between sub-grid and grid level non-local contribution should become negligible?

- Line 214-216 are a bit confusing, please explain better why this method (as opposed to the area weighting) gives different total (local?) concentrations

- line 218: non-local contributions do not have any associated emission: that is considered in the uEMEP. In general I assume they do have an associated emission. Do s refer to all source sectors in the EMEP model or only those considered in uEMEP?

- line 220/ Eq 9 is confusing to me. It should be possible to slightly rephrase the paragraph before to clarify why this needs to be done and what is done here. Also, is there an inconsistency between Eq 6 and Eq 9 regarding the source grid range, Eq 6 has I-nmw/2 . . . I+nmw/2 but here it runs from I-nmw . . . I+nmw

- Eq 10: Why the division by nsource?

- line 291: This is the first occasion that time is explicitly mentioned, worth a sentence of explanation since so far everything was stationary.

- Section 3.2: Annual mean with rotationally symmetric Gaussian plume – As the authors state, the condition of homogeneous distribution of wind speeds in all directions is typically not met. A calculation with wind roses would not add too much in complexity but would avoid this assumption.

- Line 510: traffic emissions are often described as line sources in emission inventories. What is then the appropriate uEMEP subgrid size?

- Which source sectors are included in the uEMEP for Norwegian forecasts?

- Section 5.1: Are all station types included in the validation? How different is the performance of uEMEP, does it work equally well for street canyon stations as for urban background sites? It would be interesting to indicate the station types in Fig 10a.

- Section 5.1.2: While the agreement is clearly better than with EMEP, still the correlation is quite low and there is a low bias. What is the authors' explanation, given that emissions are provided at quite high resolution? In particular for the low bias in summer, which is also seen in PM2.5 (factor 2!) – is this a regional issue (also seen in EMEP validation against background sites) or a problem in downscaling?

Technical / language

- line 66 typo: provided

- line 130 replace then with comma
- line 135 the same

- line 141 add comma after (I,J) to increase readability

- line 154 correct reference

- line 167, 176 the same

- line 250 insert comma after 'this' to increase readability

- Line 305: Define u*.

- Line 406 references missing

- line 574 reference missing

---

## Author Response (AR1)

[revised manuscript text omitted]

In addition calculations can be made on either long term mean emissions, using a rotationally symmetric dispersion kernel (Sect. 3.2), or on hourly emissions, using a slender plume Gaussian dispersion model (Sect. 3.1).

Typical source sectors downscaled using uEMEP include traffic, residential combustion, shipping and industry. The sectors addressed will depend on the availability of high resolution data for distributing them. uEMEP is only applied to the primary emissions of $PM_{10}$, $PM_{2.5}$ and $NO_x$ and does not involve any complex chemistry or secondary formation of particles. The concentrations of $NO_2$ and $O_3$ are calculated with uEMEP using a simplified chemistry scheme (Sect. 3.4 and 3.5).

**2.2 Sub-grid calculation method**

[revised manuscript text omitted]

sub-grid to the emission sub-grids. In the forecasting application discussed in Sect. 4 this requires only a 5 x 5 EMEP grid local fraction region. Sensitivity to the size of this region is discussed in Sect. S5.2. For each EMEP grid $(I,J)$ there will be an associated local fraction grid $LF(I,J,I_{lf},J_{lf})$ that specifies the fraction of the surrounding grids contributing to the $(I,J)$ grid. $I_{lf}$ and $J_{lf}$ are indexed from $-n_{lf}/2$ to $+ n_{lf}/2$.

160

With use of the local fraction then the local ($C_{G,local}$) and non-local ($C_{G,nonlocal}$) contributions from any particular primary pollutant to an EMEP grid $(I,J)$ is given by the sum of all the contributing local fraction contributions of the local sources ($s = 1$ to $n_{source}$) and the non-local contribution, specified by

165
$$C_{G,local}(I,J,I_{lf},J_{lf},s) = LF(I,J,I_{lf},J_{lf},s)\, C_G(I,J) \tag{4}$$

$$C_{G,nonlocal}(I,J) = C_G(I,J) - \left\{ \sum_{I'=-n_{mw}/2}^{+n_{mw}/2} \sum_{J'=-n_{mw}/2}^{+n_{mw}/2} \sum_{s=1}^{n_{source}} C_{G,local}(I,J,I',J',s) \right\} \tag{5}$$

Note that in Wind et al. (2020) $C_{G,local}$ and $C_G$ are termed $LP$ (local pollutant) and $TP$ (total pollutant) respectively and the local
170  fraction index is specified here using $(I_{lf},J_{lf})$ instead of $(\Delta x_s, \Delta y_s)$ This change is for compatibility with the notation used for the uEMEP application.

**2.4 Moving window calculation of local and non-local EMEP contributions**

When determining the local and non-local EMEP contribution at any uEMEP sub-grid receptor then a moving window methodology is applied. The aim of the moving window calculation is to represent as well as possible the local and non-local
175  EMEP contributions at any one sub-grid, in effect creating an EMEP grid that is centred on the receptor sub-grid. The moving window is centred on the receptor sub-grid $(i,j)$ and its size is specified by the number of EMEP grids it covers ($n_{mw}$, $n_{mw}$). The moving window region is the same as the uEMEP calculation window in extent, which is also defined by the number of sub-grids ($n_x$, $n_y$), Sect. 2.2. $n_{mw}$ is given by the user but it must not be larger than the area covered by the EMEP local fraction region ($n_{lf}$), i.e. $n_{mw} \leq n_{lf} - 1$. Fig. 1 shows an example where $n_{lf} = 5$ and $n_{mw} = 4$.

180

Since we need to account for all source contributions from EMEP within the moving window and since the sub-grids are not centred in the middle of the EMEP grids then the local contribution from the EMEP grids for any particular source sector $s$ can be written as

185
$$C_{G,local}(i,j,s) = \sum_{I'=-n_{mw}/2}^{+n_{mw}/2} \sum_{J'=-n_{mw}/2}^{+n_{mw}/2} C_{G,local}(I,J,I',J',s) \cdot w(i,j,I',J',s) \tag{6}$$

Here the weighting variable $w(i,j,I',J',s)$ refers to the weighting of the EMEP grid relative to the receptor sub-grid and the index $I,J$ refers to the EMEP grid which contains the uEMEP sub-grid $(i,j)$. For EMEP grids entirely within the moving window then this weighting will be unity, but for EMEP grids only partially within the moving window this weighting will be less than unity as part of that EMEP grid will also contribute to the non-local concentrations.

There are two methods implemented in uEMEP for specifying these weights. The simplest and most often used is area weighting where only the area fraction of the EMEP grid that is within the moving window for that particular receptor sub-grid is included in the local contribution. This is illustrated in Fig. 1 and is usually sufficient for the calculation, especially when the number of EMEP grids covered by the moving window is larger than 3 x 3. Mathematically the area weighting, $wa$, can be written as

$$wa(i,j,I',J') = \frac{\{a(i,j) \cap A(I',J')\}}{A(I',J')} \qquad (7)$$

where $A(I',J')$ is the area and position of each EMEP grid, $a(i,j)$ is the area and position of the moving window centred at the receptor sub-grid point $(i,j)$ and $a(i,j) \cap A(I',J')$ is the overlapping area of these two regions. For the case where $n_{mw} = 1$ then this area weighting is equivalent to a bilinear interpolation of the surround EMEP grids. Area weighting is not dependent on the source.

[Figure]

Schematic representation of the uEMEP moving window region

Region within which sub-grid emissions are calculated (4 x 4 EMEP grids)

Calculation sub-grid

EMEP local fraction region surrounding the calculation Sub-grid (5 x 5 EMEP grids)

Sub-grid emissions included in the uEMEP calculation region

Sub-grid emissions included in the moving window emission weighting

Sub-grid emissions not included in the uEMEP calculation region

uEMEP sub-grids

Part of EMEP local fraction grid included in the EMEP local contribution calculation

Part of EMEP local fraction grid not included in the uEMEP calculation

**Figure 1. Schematic representation of the moving window region. It shows the regions used for uEMEP calculations and the area and emission weighting selection used to determine the local and non-local EMEP contributions at the calculation (receptor) sub-grid. The extent of the sub-grids is only partially shown.**

210

When the moving window only covers a limited number of EMEP grids and when high resolution emission data is used that is compatible with the EMEP grid emissions, then this weighting can also be based on the high resolution emission data itself. This better represents the moving window concept because it reflects the effect of moving the EMEP grid to be centred on the receptor sub-grid in a more realistic way. In this case the emission weighting term (*we*) on the edge of the moving window

215 will be determined by the fraction of the total sub-grid emissions within the moving window and within the EMEP grid, instead of the area. This can be written as

$$we(i,j,I',J',s) = \frac{\sum e(i,j,I',J',s):\in\{a(i,j)\cap A(I',J')\}}{\sum e(i,j,I',J',s):\in\{A(I',J')\}} \tag{8}$$

220 where the numerator is the sum of the emissions within the intersection of *a(i,j)* and *A(I',J')* and the denominator is the sum of the emissions within *A(I',J')*. The resulting total concentration, using this method, may be higher or lower than the original EMEP concentrations because it reflects the impact of moving the EMEP grid in space. This is easiest to visualise if the moving window is the same size as the EMEP grid. If the moving window were centred on an area with concentrated emissions, that are in reality spread over two EMEP grids, then when using the emission weighting the new EMEP local contribution would

225 be higher, the non-local lower and the total would be different, see Fig. 2. The opposite is also true if the moving window were placed over a region with low emissions, the local contribution would be lower and the non-local higher. Due to this, it is not possible simply to subtract the local EMEP contribution from the total to get the non-local EMEP contribution, as detailed in Eq. 5.

230 To address this the non-local EMEP contribution is also calculated using the moving window with Eq. (9). The first term is the non-local contribution for a particular source and is calculated with the area weighting distribution since non-local contributions, those outside the local fraction region, do not have any associated emission or local fraction for weighting. An additional correction term, second term in Eq. (9), accounts for the non-local contributions from local contributions on the EMEP edge grids, those parts of the EMEP grids that are outside the moving window and not included as a local contribution

235 in Eq. 6. In those cases the local EMEP contribution outside the moving window must be converted to a non-local contribution and subtracted from the calculated non-local value, first term in Eq. 9.

$$C_{G,nonlocal}(i,j,s) = \sum_{I'=I-n_{mw}/2}^{I+n_{mw}/2} \sum_{J'=J-n_{mw}/2}^{J+n_{mw}/2} C_{G,nonlocal}(I',J',s) \cdot wa(i,j,I',J') -$$

$$\sum_{I'=-n_{mw}/2\ (I'\neq0)}^{+n_{mw}/2} \sum_{J'=-n_{mw}/2\ (J'\neq0)}^{+n_{mw}/2} \begin{pmatrix} C_{G,local}(I,J,I'\rightarrow I,J'\rightarrow J,s) \cdot w(i,j,I',J',s) \\ +C_{G,local}(I,J,I\rightarrow I',J\rightarrow J',s) \cdot w(i,j,I,J,s) \end{pmatrix} \tag{9}$$

In Eq. (9) the weighting term *w* represents either the emission (*we*) or the area (*wa*) weighting, depending on the choice of weighting method.

These local and non-local calculations are carried out for each emission source individually so the non-local contribution is

also dependent on source and the non-local component for any particular source will also contain the local contributions from the other sources. This makes creating a final non-local contribution complicated. To solve this, all the source specific $C_{G,local}$ + $C_{G,nonlocal}$ contributions are averaged and the sum of the $C_{G,local}$ source contributions are subtracted to obtain the final $C_{G,nonlocal}$. The final non-local contribution at each sub-grid $C_{SG,nonlocal}$, Eq. (1), is equivalent to the EMEP non-local $C_{G,nonlocal}$ contribution and is calculated by

$$C_{SG,nonlocal}(i,j) = \frac{1}{n_{source}} \sum_{s=1}^{n_{source}} \{(C_{G,local}(i,j,s) + C_{G,nonlocal}(i,j,s))\} - \sum_{s=1}^{n_{source}} \{C_{G,local}(i,j,s)\} \quad (10)$$

[revised manuscript text omitted]

Comparisons with EMEP NO$_2$ calculations show that this chemistry scheme matches the results obtained by EMEP over longer time periods.

**3.5 NO₂ – NOₓ conversion for annual means**

When annual mean data are used then the hourly mean formulation cannot be applied. Instead we use an empirically based conversion of $NO_x$ to $NO_2$ based on the type of formulation from Romberg (1996) and updated by Bächlin and Bösinger (2008). 3 years of Norwegian $NO_2$ measurements, 82 measurements in all, have been used to determine this relationship, Fig. 5.

$$[NO_2] = \frac{a\,[NO_x]}{[NO_x]+b} + c\,[NO_x] \qquad\qquad (38)$$

The fitted constants are determined to be a=20, b=30 and c=0.23. The estimated uncertainty in this conversion is around 10%, based on the normalized root mean square error of the fitted and observed $NO_2$ concentrations.

[Figure]

**Figure 5. NO₂ verses NOₓ annual mean concentrations for all stations in Norway in the period 2013-2015. The fitted curve is based on Eq. (38).**

This empirical relationship will vary from region to region, largely due to differences in $O_3$ concentrations and photolysis rates that are not included as part of the parameterization. If used over large regions, for example Europe, then the uncertainty in the $NO_2$ conversion will increase.

[revised manuscript text omitted]
 concentrations at each station. Shown are the results for both uEMEP and EMEP. Comparative statistics are shown for the uEMEP calculation. EMEP and observed means are also included. (b) Daily mean temporal profile averaged over all stations. Source contributions are shown for the temporal modelling results along with the EMEP 2.5 km calculation (EMEP4NO). 36 stations are used in the comparison.**

Fig. 11 shows the comparison of modelled and observed $PM_{10}$ for annual average at each station (scatter plot) and daily mean temporal profile averaged over all stations. Included in the scatter plot are the Scandinavian EMEP results at 2.5 km. The spatial correlation is low, $r^2=0.29$ for uEMEP with little negative bias (FB=-9.2%). The temporal variation over the whole year

is well represented when averaged over all stations ($r^2$=0.61) but the model has a negative bias of 4 µg/m³ over much of the summer period. Road dust events in the spring time are well captured by the emission model NORTRIP.

[Figure]

(a)

[Figure]

(b)

**Figure 11. (a) Scatter plot comparison of modelled and observed PM$_{10}$ for annual average concentrations at each station. Shown are the results for both uEMEP and EMEP. Comparative statistics are shown for the uEMEP calculation. EMEP and observed means**

**are also included. (b) Source contributions from both uEMEP and EMEP models are shown for the temporal modelling results along with the EMEP 2.5 km calculation (EMEP4NO). 45 stations are used in the comparison.**

695 Fig. 12 shows the comparison of modelled and observed PM$_{2.5}$ for annual average at each station (scatter plot) and daily mean temporal profile averaged over all stations. Included in the scatter plot are the Scandinavian EMEP results at 2.5 km. The spatial correlation is good, r$^2$=0.49 for uEMEP with little negative bias (FB=-10.5%). The temporal variation over the whole year is well represented when averaged over all stations (r$^2$=0.67) but the model has a negative bias of 2 µg/m$^3$ over much of the summer period. Residential wood combustion (heating) events in the winter are well captured by the emission model MetVed.

700

[Figure]

(a)

[Figure]

(b)

**Figure 12. (a) Scatter plot comparison of modelled and observed PM$_{2.5}$ for annual average concentrations at each station. Shown are the results for both uEMEP and EMEP. Comparative statistics are shown for the uEMEP calculation. EMEP and observed means are also included. (
[revised manuscript text omitted]

Wind, P., Rolstad Denby, B., and Gauss, M.: Local fractions – a method for the calculation of local source contributions to air
1020     pollution, illustrated by examples using the EMEP MSC-W model (rv4_33), Geosci. Model Dev., 13, 1623–1634, https://doi.org/10.5194/gmd-13-1623-2020, 2020.

1025    REVIEWER 1

Thanks to reviewer 1 for their very detailed review. The manuscript has definitely been improved because of these comments. Here follows the authors answers/comments to the review

1030

- Line 46: the Authors could mention the OSPM model as an example of a streetcanyon model to complement the overview of local scale models

* The two examples given are of urban modelling systems. OSPM is a street canyon model, not a system for modelling whole
1035    urban regions. It is part of the THOR forecast system as the last part of the cascade. We did not include it in this line but it is implicitly include via the THOR reference.

- Line 95: clarify if this option implies that emissions in the EMEP grid cell are not consistent with the ones in the sub-grid cells

1040

* This is clarified with the text. 'The independent emissions do not need to be consistent with the EMEP gridded emissions in this case.'

- Line 101: some discussion on the implications of using such inconsistent chemistry
1045    treatments in uEMEP and EMEP would be appreciated. As uEMEP is intended for applications over wide regions with significantly different chemical regimes, the simple chemistry may perform better in some environments than others

* This we have commented in Section 3.4 and 3.5 where this is discussed. In Section 3.4 we have added 'Comparisons with
1050    EMEP NO2 calculations show that this chemistry scheme matches the results obtained by EMEP over longer time periods.' and in Section 3.5 with 'This empirical relationship will vary from region to region, largely due to differences in O3 concentrations and photolysis rates that are not included as part of the parameterization. If used over large regions, for example Europe, then the uncertainty in the NO2 conversion will increase.'

1055    - Line 116 and 140: the term Csg_nonlocal(i,j) is the more complex to understand. Perhaps, an equation describing how is computed would help the reader. I appreciate the effort of the Authors to explain the method with Figure 1 and 2 and Section 2.3,

but it is still confusing how the local and non-local contributions of the EMEP grids are
used in the computation of the Csg_nonlocal term.

* We are aware that the non-local, local and the moving window concepts may not be as clear as we would like but we have
tried to explain this as best we can. These are geometrical considerations that are not easy to express in words or even equations
and best explained on whiteboards or with pen and paper. However, the reviewer has pointed out an oversite in our text. In
actual fact Csg_nonlocal (Equation 1)is equivalent to Cg_nonlocal derived in Equation 10, since it is the EMEP contribution
to the non-local subgrid. This was not explicitly mentioned but Equation 10 has now been updated, along with the text, to
indicate this.

- Line 147: More details on Wind et al. (2020) methodology would be appreciated
in the manuscript. Considering that the local fraction estimate links emissions with
concentrations, the Authors could clarify how the chemistry is treated once the tagged
emissions are dispersed in the EMEP grid cells. Are tagged primary pollutants emitted
as inert tracers or limited chemistry is considered? The details are provided in Wind
et al. (2020), but the reader would appreciate some further descriptions of the method
and limitations in the present manuscript.

1075

* We have included the following sentence concerning chemistry 'Tagged species are assumed to be inert species, primary PM
and NOX, for the downscaling application as chemical reactions are not included in the tagging.'

- Line 152: Provide which fraction of the total contribution is missed in the local fraction
estimate when using few EMEP grid cells.

* The authors perhaps did not understand how to answer this question. If all EMEP grids for the local fraction, not just 5 x 5,
were used then 100% would be included. With less grids more will be part of the EMEP non-local contribution and less a part
of the uEMEP local calculation. We attempt to address this in the sensitivity tests given in Sections 5.2 and S5.2. There we
show for example that when increasing the moving window size from 4 to 8 EMEP grids then the local contribution increases
by just 4% for PM10 and for NO2 this is 7%. There is no single answer to the reviewers question so none can be given in the
text. The reader is already referred to this sensitivity study in the text.

- Line178 Eq. 6: Why this is not divided by the sum of the weights? Following the
example in Fig. 1, you use more than 9 EMEP grid cells (adding their concentration) to
obtain the local contribution of the moving window over the i,j sub-grid cell. This results

with a local contribution overestimated somehow if nmw>1. For the case nmw=1, the
expression seems good as the sum of the weights would be 1.

1095    * The local fraction calculation from EMEP specifies in the 5 x 5 grids surrounding each grid how much that grid contributes to the central grid (Cg_local). So if all the weights (w) were 1 we would simply get the sum of all the contributions to that central grid within that area. The weighting is just to account for when a part of the grid is included in the moving window. Having read the reviewers question we see there may be some confusion concerning the notation. The terms Cg_local(I,J) refers to the contribution to any one EMEP grid from the surrounding grids. We left off this index to avoid over indexing,

1100    though this indexing is included in the Wind article. We will put this additional indexing back into the Cg_local(I,J,I_lf,J_lf) where I,J refer to the grid and I_lf, J_lf refer to the local fraction grid associated with each I,J grid and where I_lf and J_lf are indexed from -n_lf/2:n_lf/2. We thank the reviewer for this comment that, though indirectly, corrected a misunderstanding in the notation.

1105    - Line 233 Eq. 10: Why Cg(i,j,s) is divided by nsource if it is already the concentration of a specific source?

   * In Eq. 10 Cg(i,j,s) is the sum of the moving window total concentations calculated for each source, i.e. Cg(i,j,s)=Cg_local(i,j,s)+Cg_nonlocal(i,j,s). This was not specified in the paper so Equation 10 has been rewritten to reflect

1110    this. As written in the text the non-local and local contributions can be different for each source when using the emissions for weighting and each source will have a nonlocal component contributed from the other sources. So after doing this source specific calculation these must be recombined into a single nonlocal concentration. This is done in Equation 10 by taking the average of all the source specific total concentration calculations Cg(i,j,s) and then subtracting the total local contribution to get the final non-local value. This is rigourously correct when using the area weighting but is only a very close estimate when

1115    using the emission weighting. The authors realise that including the emission weighting makes this section much more complicated than otherwise required if only area weighting was used. To clarify what is being done the paragraph before Eq. 10 has been altered to read 'These local and non-local calculations are carried out for each emission source individually so the non-local contribution is also dependent on source and the non-local component for any particular source will also contain the local contributions from the other sources. This makes creating a final non-local contribution complicated. To solve this all

1120    the source specific Cg_local + Cg_nonlocal contributions are averaged and the sum of the Cg_local source contributions are subtracted to obtain the final Cg_nonlocal. The final non-local contribution at each sub-grid Csg_nonlocal, Eq.(1), is equivalent to the EMEP non-local Cg_nonlocal contribution and is calculated by '

   - Line 295: I suggest introducing in this section the meandering and traffic term described in the supplementary material. Some

1125    variables in the equations are not defined just before or after presenting the equation. It would help the reader to introduce

all the terms after the equations and specify which ones will be further described in
subsequent sections

* We have included the following paragraph in this Section 'In addition to the parameterizations presented here uEMEP also includes parameterizations, provided in the supplementary material, for plume meandering and change of wind direction (Sec. S3.4.1), traffic induced initial dispersion (Sec. S3.4.3) and road tunnel internal deposition and emissions (Sec. S3.4.5).' and have defined all variables included in these equations.

- Line 330: Mention the floor value of the wind speed imposed in the model in this part of the manuscript. Some details are only presented in the supplementary material.

* THe following sentence has been added 'A minimum wind speed of 0.5 m/s for all dispersion calculations has been imposed.'

- Line 414: a table with the sigma_init_y values per emission source would be appreciated.

* The sigma_init_y, as mentioned in the text, is defined almost exclusively by the grid size, rather than the physical process but we have included the additional text 'traffic and 5 m for shipping, heating and industry'

- Line 518: an order of magnitude of the maximum distance allowed in the dispersion of the Gaussian model would be appreciated (i.e., 250 m).

* It is not clear to the authors what the reviewer is refering to here as there is no mention of this in this line. We do not know which 'maximum distance allowed' the reviewer is referring to. The distance the plume can travel is defined by the size of the moving window, if that was what is meant here. If the reviewer is referring to the sub-grid size then there is no numerical limit but we have never applied the model to a larger sub-grid than 500 m.

- Line 583: Is ozone also a product used from uEMEP? Is there any evaluation done for this pollutant?

* Ozone is a product and this is also assessed but we simply did not include it here. There are very few ozone stations in Norway where the model was assessed. This link shows extensive evaluation, also for ozone, but is in Norwegian (https://www.met.no/prosjekter/luftkvalitet/evaluering-av-luftkvalitets-modellen)

- Line 631: Some discussion about the improvement in the daily cycle of the uEMEP

1160    results compared with EMEP would be appreciated. Local models use to improve the

traffic peaks but also may inherit issues with the temporal profiles and the boundary

layer evolution. The validation section could be improved introducing some discrimination between types of sites (rural,

industrial, suburban, urban). I suggest presenting all

the material of subsections 5.1.1, 5.1.2 and 5.1.3 under section 5.1 as those sections

1165    consist only in a single paragraph.

* We have reduced Section 5.1 to a single section, as suggested by the reviewer. As listed there are a limitted number of

monitoring sites in Norway, with 90% being traffic stations. There is 1 urban site, 3 suburban sites, 2 rural sites and 1 industrial

site. This lack of representation does not warrant individual selection and presentation. The EMEP model run in Norway uses

1170    the same emission data as the local scale uEMEP, only aggregated to grids. So the traffic variation is exactly the same, if this

is what the reviewer is refering to. In general we have tried to keep the validation to a minimum as this will be more thoroughly

assessed at a later date. The validation is intended to show that the model works, rather than a detailed analysis. The paper is

intended as a model description primarily.

1175    - Line 653: What missing processes could explain the remaining bias during the summer period in both PM10 and PM2.5?

* We believe a large part is secondary organics, but that is currently just speculation so this was not taken up in the article. We

intend a more detailed evaluation of many more years in a later article.

1180    - Line 700: it is counter-intuitive having more non-local EMEP contributions with smaller

moving windows. Could the Authors clarify this in the text? If less EMEP grid cells are

used in the moving window, less non-local contributions would be expected.

* Non-local contributions come from outside the moving window. The larger the moving window the less comes from outside

1185    so this is intuitively correct. Said differently, the larger the moving window the more local contribution as well. This is decribed

in Section S5.2.

- Line 796: There are still some street-canyon processes that uEMEP cannot represent,

particularly in compact cities with high street aspect ratios. The Authors should mention

1190    this in this last concluding remark.

* We have added the text to this line 'It can also represent concentrations down to street level, though not street canyons, '. It

was pointed out before that it is not obstacle resolving but it does not hurt to mention this a second time.

1195 - Line 29: the acronym CTM is used several times in the manuscript but defined in
Line 71. Please, define the acronym already in the introduction and use directly the
acronym in the rest of the manuscript.

* Done

1200

- Line 51: use coma instead of a semi-colon in the reference

* Done

1205 - Line 58: the reference Wind et al. (2020) is not provided in the reference section.

* That was strange, but inserted.

- Line 154: fix the Section number. Here and in other parts of the manuscript, the
1210 number of the reference to specific sections is 0.

* Due to automatic reference system that stopped working. This is now fixed

- Line 245: Use Eq. instead of Equ. in the Figure caption.

1215

* Done

- Line 362: Monin–Obukhov is mistyped in different parts of the manuscript.

1220 * Done

- Line 362: the Monin-Obukhov length and the surface roughness have already been
used before in the manuscript. Define them there only once.

1225 * Done

- Line 371 Table1: please, use consistent notation for the boundary layer height and

Monin-Obukhov length. Both have been introduced before as H and L

* Done

- Line 407 and 574: fix the section number that appears in the reference Sect. 0.

* Done

- Line 646: the statistics presented in panel (a) should be introduced in the caption
specifying for which model are computed. In panel (b), the Authors could remove the
shipping and industry labels in the legend as no information is shown in the figure.

* The contribution from industry and shipping is present but very small, in this case. Since these were calculated we will keep
them in the legend. We have added in the figure caption that the statistics in panel (a) refer to the uEMEP model and we have
reorganised the statistics text in the plot itself to better reflect this.

- Line 661: There is too much information in Figure 11. I suggest presenting the nonlocal contribution of EMEP and not the
detailed composition of it. Though of interest,
it is impossible to appreciate EMEP4NO line and some artefacts appear as the white
contribution above EMEP PRIMARY blue fraction.

* We agree that the plot was very busy and we have now aggregated all species into the non-local EMEP, as suggested by the
reviewer.

- Line 679: avoid using subsections that consist of a single paragraph.

* The authors used this form to have consistent cross referencing to the supplementary material but understand it would seem
a little strange in this context. We have removed the numbering but have kept the headings to delineate between the different
sensitivity studies.

- Line 719: to be consistent with the supplementary material the coefficient of determination of the station mean time series of
uEMEP should be 0.79, not 0.80. Harmonise
the number in both documents

* Done

- Line 728: I suggest merging Sections 6 and 7.

1265

* The authors would like to keep these as two seperate sections, as is often the case for discussion and conclusion.

- Line 13: Use section S1 instead of S3 and number accordingly the rest

1270 * As explained in the text we use this numbering so it is easy to cross reference between supplementary material and the main document. Given the nature of the supplementary material, that it provides extra details on particular sections in the main document, the authors feel this method of numbering is more appropriate and will keep it. This is already stated at the start of the document.

1275 - Line 106: It should be Eq. (15a).

* Done

- Line 242: Why the inverse of the wind speed is used instead of wind speed?

1280

* In dispersion calculations it is the inverse wind speed that is multiplied with emissions and the dispersion intensity to determine the concentration, Eq. 11. When averaging then it is the mean of the inverse of the wind speed that should be used, rather than the mean of the wind speed.

1285 - Line 314: In the figure caption, it should be Fig. S4 instead of S2.

* Done

- Line 328: The observation measurement could be provided in Fig. S6.

1290

* The comparison with observations has already been made in Figure 10a in the main paper. Since the major aim of this sensitivity study was to assess the dependence on resolution the authors do not think it is necessary to repeat that comparison here, in an already crowded plot. We do not include the observations again here.

1295 - Line 368: Why Figure S8a is different from Figure 10b? The caption describes the

same results.

* The reviewers are correct that these two plots should be the same. This has now been fixed. The discrepency was due to the way concurrent measurements and model results were selected. The reviewers will also note that the mean of scatter plots is
1300 not always the same as the mean of the time series. This is also due to the selection criteria for annual means requiring 75% coverage per station whilst daily mean plots of the average of all stations do not have the same requirement.

- Line 385: A value of 0.1 would likely provide an even closer fit to observations.

1305 * A lower value than 0.15 would probably give a better fit but this was not assessed as it lay outside the expected NO2/NOx emission ratio range for vehicles in Norway.
* * *
1310 REVIEWER 2
* * *
Thanks to reviewer 2 for their comments and time. Some aspects of the modelling, particularly how the local window local/non-local works are difficult to explain but I have tried to improve on this. Reviewer 1 also commented on this. Here follows the
1315 answers to the reviewers questions and improvements.

- line 33 / "near street level modelling": What is then the ambition of the model? Is
it supposed to represent concentrations at roadside monitoring sites or background
sites?

1320

* It will represent concentrations at roadside monitoring sites, and the validation for NO2 confirms this, when using a resolution of 25 m. Even so the sensitivity tests to resolution, Section S5.3, show that good results are still obtained at 100 m. It does not however well represent street canyon sites as the Gaussian model used has no obstacles. One would then expect an underestimate at these sites.

1325

- What is the meaning of resolution <100m when there is no local topography modelling involved? Wouldn't building layout, air flows in the street canyon etc need to be
accounted for at these very local scales?

1330  * Without including obstacles the increased resolution allows the concentration gradients at roadside to be better described. The reviewer is right to point out that, if this was to be done properly at < 100 m, then buldings need to be included. However, uEMEP is intended for application over country scales and that level of detail is not achievable.

- line 85: Which source sectors are included in the uEMEP downscaling calculations?
1335  Traffic, residential, any other? Should be mentioned somewhere in Sect 2.1

* The downscaled sectors depend on the application so this is not expllicitly named until the application is defined in Section 4.2. However, we have included the following text in Section 2.1, line 100 'Typical source sectors downscaled using uEMEP include traffic, residential combustion, shipping and industry. The sectors addressed will depend on the availability of high
1340  resolution data for distributing them'

- line 150. "neighbour cells" sounds as if only +/- 1 in each direction but I understand from the next sentence that the local fraction region can be quite large. Please clarify in the text.
1345
* We have changed that sentence to read 'The local fraction region extent is then limited.'

- line 153. Perhaps I missed it but it would be good to have a paragraph somewhere that explains the difference between the different domains (uEMEP vs local fraction vs
1350  moving window) as it is a bit confusing to the reader

* Moving window and EMEP local fraction region are described in the text seperately but these are also visualised in Fig. 1. That was indeed the intention of Fig. 1. We believe this is sufficient explanation.

1355  - Sect 2.3-2.4: These sections are difficult to follow, I would suggest restructuring 2.3 and 2.4 into one (The second sentence of 2.3 already refers to 2.4)

* We would prefer to keep these as two different sections. The first (2.3) applies to the local fraction calculation from EMEP and the second (2.4) the moving window calculation in uEMEP. Though the second utilises the first, they are two distinct
1360  calculations. Reviewer 1 also commented on this section and as a result additional text and a change in formulation of the equations have been implemented. We belive this has helped to clarify these sections.

- Sect 2.4: This is rather complicated to follow for an effect that is probably secondorder. How much is gained by the complicated moving window calculation of nonlocal contributions at sub-grid resolution? With a reasonably big local fraction tracking

domain, the difference between sub-grid and grid level non-local contribution should

become negligible?

* The reviewer is correct, first that it is complicated and second that it would not matter if the local fraction domain and moving window were sufficiently large. This is also stated in the text. However, there will always be an edge somewhere to the moving window domain and we consider it necessary to implement a method that can deal with this limit properly, especially when just 1 large EMEP grid, for example 15 km, is used.

- Line 214-216 are a bit confusing, please explain better why this method (as opposed to the area weighting) gives different total (local?) concentrations

* We have tried to clarify this in the text with an extended explanation 'The resulting total concentration, using this method, may be higher or lower than the original EMEP concentrations because it reflects the impact of moving the EMEP grid in space. This is easiest to visualise if the moving window is the same size as the EMEP grid. If the moving window were centred on an area with concentrated emissions, that are in reality spread over two EMEP grids, then when using the emission weighting the new EMEP local contribution would be higher, the non-local lower and the total would be different, see Fig. 2. The opposite is also true if the moving window were placed over a region with low emissions, the local contribution would be lower and the non-local higher. Due to this, it is not possible simply to subtract the local EMEP contribution from the total to get the non-local EMEP contribution, as detailed in Eq. 5.'

- line 218: non-local contributions do not have any associated emission: that is considered in the uEMEP. In general I assume they do have an associated emission. Do s

refer to all source sectors in the EMEP model or only those considered in uEMEP?

* We are refering only to the sources that are downscaled using uEMEP but we have reworded to make this clear. 'The first term is the non-local contribution for a particular source and is calculated with the area weighting distribution since non-local contributions, those outside the local fraction region, do not have any associated emission or local fraction for weighting.'

- line 220/ Eq 9 is confusing to me. It should be possible to slightly rephrase the

paragraph before to clarify why this needs to be done and what is done here. Also, is

there an inconsistency between Eq 6 and Eq 9 regarding the source grid range, Eq 6

has I-nmw/2 . . . I+nmw/2 but here it runs from I-nmw . . . I+nmw

\* The reviewer is correct, there is an inconsistency between Eq. 6 and 9. Thank you pointing this out. We have corrected Eq.
9. Reviewer 1 also commented here and some additional updates of the Equation indexing has been made. We have rephrased this paragraph to read ' An additional correction term, second term in Eq. (9), accounts for the non-local contributions from local contributions on the EMEP edge grids, those parts of the EMEP grids that are outside the moving window and not included as a local contribution in Eq. 6. In those cases the local EMEP contribution outside the moving window must be converted to a non-local contribution and subtracted from the calculated non-local value, first term in Eq. 9.'. These are geometrical arguements that are difficult to explain with words and equations but we hope the concept has become clearer.

- Eq 10: Why the division by nsource?

\* This was also commented by reviewer 1 and we have rewritten the text and reformulated Eq. 10 to make this clearer. The term $n\_source$ was used to average $CG(i,j,s)$ since this value contains non-local contributions from other sources as well, each source having it's own local and non-local contribution. In addition the term $CG(i,j,s)$, in Eq. 10, was actually never defined (it is $CG(i,j,s) = CG,local(i,j,s) + CG,nonlocal(i,j,s)$). The text now reads,  'These local and non-local calculations are carried out for each emission source individually so the non-local contribution is also dependent on source and the non-local component for any particular source will also contain the local contributions from the other sources. This makes creating a final non-local contribution complicated. To solve this, all the source specific $CG,local$ + $CG,nonlocal$ contributions are averaged and the sum of the $CG,local$ source contributions are subtracted to obtain the final $CG,nonlocal$. The final non-local contribution at each sub-grid $CSG,nonlocal$ , Eq. (1), is equivalent to the EMEP non-local $CG,nonlocal$ contribution and is calculated by ...'

- line 291: This is the first occasion that time is explicitly mentioned, worth a sentence
of explanation since so far everything was stationary.

\* We have made this clearer by writing 'pollutant travel time (t) from source'.

- Section 3.2: Annual mean with rotationally symmetric Gaussian plume – As the authors state, the condition of homogeneous distribution of wind speeds in all directions
is typically not met. A calculation with wind roses would not add too much in complexity
but would avoid this assumption

1430    * Yes, wind rose calculations could be made but this does require use of the hourly meteorological data at every point in space within the model domain. This is much more complicated and time consuming than this simple analytical methodology. Comparisons with hourly calculations, Section S5.1, show the assumption works quite well.

- Line 510: traffic emissions are often described as line sources in emission inventories.

1435    What is then the appropriate uEMEP subgrid size?

* We find 25 m is sufficient resolution (around the width of a multi-laned road) and little is gained by higher, or even slightly lower resolutions, see Section S5.3. We use 25 m for receptor calculations though 25 m is prohibitive for large scale map making.

1440

- Which source sectors are included in the uEMEP for Norwegian forecasts?

* This is stated in Section 4.2

1445    - Section 5.1: Are all station types included in the validation? How different is the

performance of uEMEP, does it work equally well for street canyon stations as for urban

background sites? It would be interesting to indicate the station types in Fig 10a.

* All stations are included. In Norway there are very few traffic stations that could be called 'street canyon', around 3 of these.

1450    The rest are in fairly open road situations. There are also very few urban background sites, around 3 of these as well. Since this article is primarily a model description we tried to reduce the validation to a minimum. We believe that a more detailed assessment is more appropriate for a seperate paper which includes many more years of data and a more thorough investigation. For the moment the authors think the current validation is sufficient.

1455    - Section 5.1.2: While the agreement is clearly better than with EMEP, still the correlation is quite low and there is a low bias. What is the authors' explanation, given

that emissions are provided at quite high resolution? In particular for the low bias in

summer, which is also seen in PM2.5 (factor 2!) – is this a regional issue (also seen in

EMEP validation against background sites) or a problem in downscaling?

1460

* Without having direct proof we believe the low values in the summer are due to too low estimates of secondary organics in EMEP and not a problem with local sources. This is being looked at. The spatial correlation for PM2.5 annual mean is $r^2=0.49$, which is significantly better than EMEP and given the complexities of PM emissions and processes a reasonable result. Low

correlation for PM10 is dependent on all the same PM2.5 uncertainties but in addition, in Norway and other Scandinavian countries, road dust emissions have a significant impact on PM10. This emission source is very difficult to model, though we do use the NORTRIP road dust model for this which is currently state of the science. As mentioned in the previous answer we will be investigating these problems in a later study that concentrates on the results rather than the model.

- line 66 typo: provided

* Done

- line 130 replace then with comma

* Done

- line 135 the same

* Done

- line 141 add comma after (I,J) to increase readability

* Have added a comma on both sides, I think that is correct.

- line 154 correct reference

* Done

- line 167, 176 the same

* Cannot find this line reference

- line 250 insert comma after 'this' to increase readability

* Done

- Line 305: Define u*.

* Done 'friction velocity'

1500

- Line 406 references missing

* Done

1505    - line 574 reference missing

* Done

1510

1515

1520

1525